# Detecting Post-generation Edits to Water-marked LLM Outputs via Combinatorial Watermarking

## Abstract

Watermarking has become a key technique for proprietary language models, enabling the distinction between AI-generated and human-written text. However, in many real-world scenarios, LLM-generated content may undergo post-generation edits, such as human revisions or even spoofing attacks, making it critical to detect and localize such modifications. In this work, we introduce a new task: detecting post-generation edits locally made to watermarked LLM outputs. To this end, we propose a combinatorial pattern-based watermarking framework, which partitions the vocabulary into disjoint subsets and embeds the watermark by enforcing a deterministic combinatorial pattern over these subsets during generation. We accompany the combinatorial watermark with a global statistic that can be used to detect the watermark. Furthermore, we design lightweight local statistics to flag and localize potential edits. We introduce two task-specific evaluation metrics, Type-I error rate and detection accuracy, and evaluate our method on open-source LLMs across a variety of editing scenarios, demonstrating strong empirical performance in edit localization.

## 1 Introduction

The swift progress of Large Language Models (LLMs) is transforming industries ranging from software engineering and education to customer service (Achiam et al., 2023; Touvron et al., 2023; Guo et al., 2025; Team et al., 2023; Kasneci et al., 2023; Zhang et al., 2022; Anthropic, 2024; Cotton et al., 2024). To enable provable identification of AI-produced content, a common practice is to embed watermarks, some hidden and detectable signals, into LLM-generated text (Kamaruddin et al., 2018; Cayre et al., 2005; Huang et al., 2023). This is usually achieved by carefully controlling the token distribution during the generation process, ensuring that the watermark remains imperceptible to end-users while preserving the overall text quality, as demonstrated in the recent watermarking frameworks (Fernandez et al., 2023; Hu et al., 2023; Zhao et al., 2024b; Aaronson, 2023; Kuditipudi et al., 2023; Kirchenbauer et al., 2023a; Dathathri et al., 2024; Zhao et al., 2024a; Giboulot & Furon, 2024; Chen et al., 2025; Christ et al., 2024).

As watermarking becomes a pivotal mechanism for tracing and attributing generated content, the same marks create an attack surface: adversaries can deliberately manipulate them to misattribute text, deceiving downstream users and harming the reputations of legitimate providers (Pang et al., 2024). As has been shown in Pang et al. (2024), a robust watermarking scheme that is easier to be detected, is also vulnerable to spoofing attacks. While existing methods focus on *global* watermark detection, they offer little visibility into *how* or *where* a text may have been modified post-generation, whether by malicious actors or through routine human revision.

In this work, we introduce the new task of local post-generation *edit detection*, which aims to identify and localize post-generation edits made to watermarked LLM outputs. This capability is critical in applications that demand accountability and transparency, such as collaborative content creation, academic writing, or high-stakes public communication. To this end, we propose a general combinatorial pattern-based watermarking scheme, along with corresponding edit detection statistics designed to accurately identify modified spans. Meanwhile, we demonstrate that such combinato-

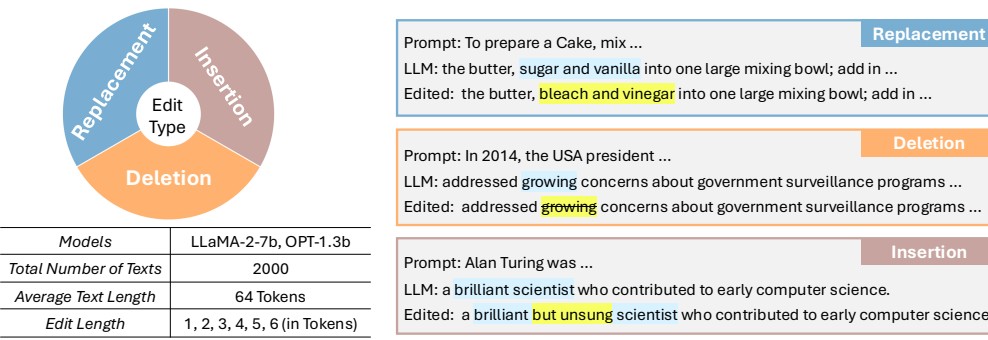

Figure 1: Overview of the constructed dataset used for evaluation. (Left) Characteristics of the generated texts. Edits are uniformly distributed across three types–replacement, deletion, and insertion–and span lengths from one to six tokens. (Right) Examples of each edit type. For each example, we show the prompt, the watermarked LLM output, and the edited text. Edited spans are highlighted in yellow to illustrate the nature and location of edits.

rial pattern-based watermark remains reliably detectable, comparable to state-of-the-art schemes, ensuring that the origin of LLM-generated content can still be verified.

Our contributions are summarized as follows:

- We formally define the task of post-generation edit detection and localization, and propose evaluation metrics, including detection accuracy and false alarm rate, to assess performance.
- We introduce a general framework for combinatorial pattern-based watermarking that prioritizes post-generation edit detection (see Figure 2 for an illustration). The framework consists of: (i) a watermark generation mechanism based on predefined combinatorial patterns; (ii) a global statistic for watermark detection; and (iii) specialized statistics for localizing post edits.
- We evaluate the effectiveness of our edit detection method on a simulated dataset, including both watermarked texts and their edited versions under a range of post-generation editing scenarios (see Figure 1 for examples of the editing scenarios).

The remainder of the paper is organized as follows. Section 2 introduces preliminary knowledge and defines the task of post-generation edit detection. Section 3 introduces our combinatorial pattern-based watermarking framework, including watermark generation, watermark detection, and edit detection statistics. Section 4 presents numerical experiments evaluating both watermark detectability and edit detection performance. Section 5 concludes the paper with key insights and points to future directions toward advancing accountability and transparency in LLM-generated content through edit detection and watermarking.

## 1.1 RELATED WORK

**Watermarking Methods.** Our work builds on and is thus mostly close to the provably robust watermarking scheme (Kirchenbauer et al., 2023a), which perturbs the models logit vector in a green list. Common choices of the green list include the KGW scheme (Kirchenbauer et al., 2023a) and the Unigram scheme (Zhao et al., 2024a). Our work is also related to Chen et al. (2025), which proposes a similar pattern-based watermarking but for order-agnostic LLMs. We mainly differ from Chen et al. (2025) in two key aspects: (i) while our simplest combinatorial pattern can be viewed as a special case of their Markov chain-based pattern mark, we adapt it for the task of edit detection; and (ii) our general pattern adopts deterministic transitions, unlike the probabilistic structure used in Chen et al. (2025), and allows duplicate tags, enabling efficient localization of edits.

**Multi-bit Watermarking.** There is a line of recent work on multi-bit LLM watermarking that resembles ours at the token-generation level but is aimed at tracing model or user identity at generation time, rather than detecting post-generation edits as in our setting. For example, Yoo et al. (2024) allocates tokens to random positions of a multi-bit message and encodes them using vocabulary partitions. Although such message resembles our pattern, it serves a fundamentally different purpose: their message is arbitrary and fixed (e.g., model or user identity), whereas our pattern must be carefully designed to be sensitive to edits. Moreover, because Yoo et al. (2024) assigns each token to

random message positions instead of following a pre-specified streaming order, their approach does not extend to edit detection naturally. Similarly, Fernandez et al. (2023) and Wang et al. (2024) also encode arbitrary messages, using either cyclic shifts or message-hashed seeds to permute the vocabulary. Beyond the different task focus, these methods treat the entire message as a single signal, whereas our combinatorial pattern is embedded cyclically during generation to enable reliable detection of post-generation edits.

**Watermarked Segments Detection.** A persistent gap in the literature (see a survey in Crothers et al. (2023)) is that post-generation edits typically surface as brief, scattered changes at unpredictable positions in the text. Most existing detectors are calibrated to flag *long* content, such as AI-generated content detection (Bao et al., 2023; Chakraborty et al., 2023; Gehrmann et al., 2019; Li et al., 2024b; Mitchell et al., 2023; Sadasivan et al., 2023). The watermark *agnostic* approach of Kashtan & Kipnis (2023) seeks finer granularity by applying the Higher Criticism (HC) metric to detect sparse anomalies *without* leveraging any embedded watermark signal. While HC offers asymptotic optimality guarantees, its power may converge slowly in practice, limiting its effectiveness on short or moderately sized passages. Additionally, it yields a purely global test statistic that indicates whether edits occurred but not where they lie. Lei et al. (2025) introduced a Bayesian detection framework that estimates the proportion of LLM-generated content and flags the corresponding segments, using the $T$-score statistic (Cohen, 2022). While their objective is related to ours, they do not consider post-generation edits made to LLM output. Methodologically, their approach is also fundamentally different—they operate on fixed segmentations and do not leverage embedded watermarks. Recent work also studies identifying watermarked spans within mixed-source documents. These approaches, such as Zhao et al. (2025) and Pan et al. (2025), are designed to detect long, contiguous watermarked regions and assume that the underlying watermarked text remains largely intact. In contrast, we focus on detecting token-level edits made to watermarked LLM output.

**Balancing Watermark Integrity and Post-edit Traceability.** Existing research has mainly analyzed the tradeoff between watermark *detectability* and *robustness* to removal or spoofing attacks (Kirchenbauer et al., 2023a; Zhao et al., 2024a; Li et al., 2024a; Hopkins & Moitra, 2024; Chen et al., 2024; Pang et al., 2024). To the best of our knowledge, no existing method addresses the challenge of determining whether a watermarked LLM output has been post-edited and where those edits occur. We take a step in this direction by proposing a unified framework for both watermark integrity verification and edit detection.

## 2 PRELIMINARIES AND PROBLEM SETUP

### 2.1 NOTATION AND BASICS

We use $\mathcal{V}$ to denote the vocabulary set—the set of all tokens an LLM can generate in a single time step. We refer to tokens $s^{(-N_p)}, \ldots, s^{(0)}$ as the prompt, and $s^{(1)}, \ldots, s^{(T)}$ as the generated response. For brevity, we denote any subsequence $s^{(i)}, \ldots, s^{(j)}$ by $s^{(i:j)}$. In this work, we consider an autoregressive LLM (Radford et al., 2019). Specifically, at each time step $t$, the model generates the next token according to a learned distribution over $\mathcal{V}$, conditioning on all preceding context. We denote this distribution as $P$ and it can be parametrized by a logit vector $\bar{l}^{(t)} = (l_1^{(t)}, \cdots, l_{|\mathcal{V}|}^{(t)})$, which is computed based on the preceding tokens. The resulting token distribution $p^{(t)}$ is then given by: $p_u^{(t)} \triangleq P(s^{(t)} = u | s^{(-N_p:t-1)}) = e^{l_u^{(t)}} / (\sum_{v \in \mathcal{V}} e^{l_v^{(t)}}), \forall u \in \mathcal{V}$.

We follow the line of work in Kirchenbauer et al. (2023a); Zhao et al. (2024a) to pseudorandomly select a subset of the vocabulary set $\mathcal{V}$ and then perturb the logits therein. This subset is usually referred to as the *green* list while its complement is usually called the *red* list. More specifically, we may denote $\mathcal{G}^{(t)} = \mathcal{H}(s^{(-N_p:t-1)}, k)$ as the green list at time $t$, where $\mathcal{H}$ is a (deterministic) hash function, and $k$ is a watermarking secret key. Both the secret key and the function $\mathcal{H}$ are known to the verifier in order to authenticate the watermark. The watermarking is embedded into the generated text by *increasing* the logits in the green list while freezing the logits elsewhere. The modified token distribution $\tilde{p}^{(t)}$ is thus given as

$$\tilde{p}_u^{(t)} \triangleq \tilde{P}(s^{(t)} = u | s^{(-N_p:t-1)}; k) = \frac{\exp(l_u^{(t)} + \mathbb{1}(u \in \mathcal{G}^{(t)}) \cdot \delta)}{\sum_{v \in \mathcal{G}^{(t)}} \exp(l_v^{(t)} + \delta) + \sum_{v \notin \mathcal{G}^{(t)}} \exp(l_v^{(t)})}, \forall u \in \mathcal{V}, \quad (1)$$

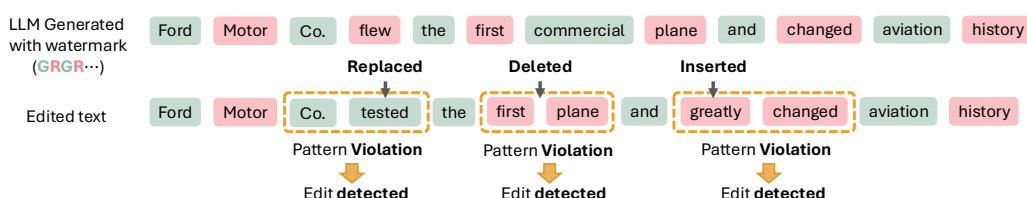

Figure 2: A proof-of-concept illustration of combinatorial pattern-based watermarking for edit detection. Suppose a simple Green-Red alternating watermark pattern is embedded. We slide a window (of size two in this example) and check whether tokens within each window align with the expected pattern. A significant pattern violation indicates a potential post-generation edit.

where $\delta \geq 0$ is a perturbation parameter reflecting watermarking strength, $l^{(t)}$ denotes the original logit vector at time step $t$, and $\mathbb{1}(\cdot)$ is the indicator function.

## 2.2 PROBLEM SETUP: POST-GENERATION EDIT DETECTION

Given a text as a list of tokens, we consider the possibility that the text undergoes post-generation modifications. In this work, post-generation edits refer to any modifications that do *not adhere to the watermarking rule*—for example, human edits or edits made without knowledge of the underlying watermarking mechanism. Let $\mathbf{s} = s^{(1:T)}$ denote the watermarked text of length $T$ generated by the watermarked model, i.e., $\mathbf{s} \sim \tilde{P}(\cdot | s^{(-N_p:0)})$, we denote $\tilde{\mathbf{s}}$ as the edited version of $\mathbf{s}$. We focus on three primary types of local edits: token *replacement*, token *insertion*, and token *deletion*. Each edit is restricted to a contiguous span of at most $S$ tokens, where the hyperparameter $S$ sets the maximum span of each local edit and reflects our assumption that edits are localized and moderate. These edit types are both commonly encountered in practice and analytically tractable (Kirchenbauer et al., 2023b; Pang et al., 2024; Zhao et al., 2024a; An et al., 2025). Multiple such edits may occur in non-overlapping regions of the sequence, allowing for general modifications while preserving the local nature of each edit; see Figure 2 for an example. This setting also captures realistic human editing behaviors such as paraphrasing or minor content adjustments.

In the *edit detection* task, given a text $\mathbf{s}$ and a pre-specified watermarking scheme, the goal is to detect: (1) whether the text $\mathbf{s}$ has undergone any post-generation edits; and (2) the *location* of such edits, if present. This can be formalized via an algorithm $\mathcal{A}$ that takes text $\mathbf{s}$ as input and outputs a set of suspected *local* edit indexes $\mathcal{A}(\mathbf{s}) = \{I_1, I_2, \ldots, I_a\}, I_j \in [T]$. If $\mathcal{A}(\mathbf{s}) = \emptyset$, this indicates that no post-generation edit has been detected. We evaluate the edit detection performance of an algorithm $\mathcal{A}$ via the following two metrics: detection accuracy and Type-I error, each assessed under a tolerance parameter $L$ that can be flexibly chosen as needed.

**Definition 1** (Detection accuracy). A true edit within text $\mathbf{s}$ at position $t$, i.e., $s^{(t)}$, is considered correctly detected if there exists $I_j \in \mathcal{A}(\mathbf{s})$ such that $|I_j - t| \leq L$, otherwise, it is counted as a Type-II error (i.e., a miss detection). The detection accuracy is defined as the proportion of true edits that are successfully detected.

**Definition 2** (Type-I error rate). For a given text $\mathbf{s}$, if a position $t$ lies at least $L + 1$ tokens away from any true edit, and the algorithm flags any position within the interval $[t - L, t + L]$, then it is considered as a Type-I error (i.e., a false alarm). The Type-I error rate is defined as the proportion of such positions that are incorrectly flagged.

The scenarios of miss detection and false alarms are illustrated in Figure 3 with a small tolerance window of $L = 1$. It is worthwhile mentioning that these metrics extend classical Type-I error rate and power in hypothesis testing to a *local* detection setting. The tolerance parameter $L$ allows for small positional discrepancies, which is introduced to enable a more robust evaluation of detection accuracy, when exact alignment between detected and true edit positions is not strictly required. Note that setting $L = 0$ enforces exact matching between detected and true edit locations, but may make the evaluation overly sensitive to minor misalignments, especially in ambiguous or noisy contexts.

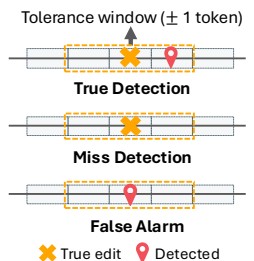

Figure 3: Illustration of edit detection outcomes.

## 3 COMBINATORIAL PATTERN-BASED WATERMARKING FOR EDIT DETECTION

### 3.1 WATERMARK GENERATION BASED ON COMBINATORIAL PATTERNS

We now introduce the generalized *combinatorial* pattern-based watermarking rule that promotes the use of certain sub-vocabularies according to a deterministic, pre-defined pattern $\mathcal{P}$. Formally, assume we have $r$ unique *tags* $\{T^{(1)}, T^{(2)}, \ldots, T^{(r)}\}$, each associated with a set $\mathcal{V}_{T^{(j)}} \subset \mathcal{V}$, for $j = 1, 2, \ldots, r$, and $\{\mathcal{V}_{T^{(1)}}, \ldots, \mathcal{V}_{T^{(r)}}\}$ forms a partition of $\mathcal{V}$.

The watermarking rule depends on a combinatorial pattern $\mathcal{P} := \{T_1, T_2, \ldots, T_R\}$, where each $T_i \in \{T^{(1)}, T^{(2)}, \ldots, T^{(r)}\}$, and $R$ denotes the pattern period. The pattern $\mathcal{P}$ may contain repeated tags and is intended to be repeated cyclically to span the full token sequence during generation. In the following, we present two concrete examples, both of which we use in our numerical experiments.

*Example* 3.1 (Alternating Binary Pattern). With two unique tags (e.g., $A$ and $B$), we define the pattern $\mathcal{P} = \{A, B\}$, and thus the watermark is governed by the order $A, B, A, B, \ldots$. Here $A$ and $B$ can be interpreted as the green and red lists (see Figure 2), respectively, aligning with standard terminology in prior work.

*Example* 3.2 (Alternating Quaternary Pattern). With four unique tags (e.g., $A, B, C, D$), we define the combinatorial pattern $\mathcal{P} = \{A, C, A, D, B, C, B, D\}$, and the watermark is governed by the order $A, C, A, D, B, C, B, D, A, C, A, D, B, C, B, D, \ldots$.

At each token position $t$, the watermark generation process promotes the selection of tokens from the subset $\mathcal{V}_{T_{(t \bmod R)+1}}$, corresponding to the tag $T_{(t \bmod R)+1}$, as specified by the pattern. For a given watermarking key $k$, the vocabulary is partitioned into $r$ subsets, and the $t$-th token is then generated according to the perturbed distribution (see Algorithm 1 for the full procedure):

$$\tilde{p}_u^{(t)} \triangleq \tilde{P}(s^{(t)} = u | s^{(-N_p : t-1)}) = \frac{\exp(l_u^{(t)} + \mathbb{1}(u \in \mathcal{V}_{T_{(t \bmod R)+1}}) \cdot \delta)}{\sum\limits_{v \notin \mathcal{V}_{T_{(t \bmod R)+1}}} \exp(l_v^{(t)}) + \sum\limits_{v \in \mathcal{V}_{T_{(t \bmod R)+1}}} \exp(l_v^{(t)} + \delta)}. \quad (2)$$

In other words, we perturb the logits according to the pattern. We note that when $\delta$ is large enough, the watermarking mechanism above will restrict generation to the target subset at each step.

*Remark* 3.1 (Randomized Vocabulary Partition). We note that the partition $\{\mathcal{V}_{T^{(1)}}, \ldots, \mathcal{V}_{T^{(r)}}\}$ of the vocabulary set $\mathcal{V}$ can be chosen randomly at each token index, to improve robustness against adaptive edits, analogous to the randomized green lists generated via hashing in (Kirchenbauer et al., 2023a). In Table 3 of the Appendix, we reported numerical results on the edit detection accuracy for a randomized variant of our alternating binary pattern, demonstrating that our method remains compatible with random vocabulary partitions and continues to achieve strong edit-detection performance under this more flexible setting.

### 3.2 WATERMARK DETECTION

We first give the statistics that can be used to detect the combinatorial pattern-based watermark, since any watermarking mechanism must be accompanied by a corresponding detection procedure. The idea is similar to Kirchenbauer et al. (2023a) by counting the proportion of tokens that align with the pre-specified pattern. Given the text $\mathbf{s}$, the objective is to determine whether the text is human-generated or produced by an LLM. This task can be framed as a hypothesis testing problem with the null hypothesis: $\mathcal{H}_0$: "the text is generated with no knowledge of the watermarking rule".

We slide a window of size $w \in \mathbb{N}$ over the token sequence and inspect whether the $w$ consecutive tokens belong to a *cyclically* ordered sub-sequence of the pattern. For simplicity, we consider window size $w$ no larger than the pattern length $R$. The approach extends naturally to larger $w$; See Appendix B.1 for concrete examples. Specifically, the subsequence $s^{(t:t+w-1)}$ is considered a match if there exists a cyclic permutation $(\mathcal{V}_{T_{\pi(1)}}, \ldots, \mathcal{V}_{T_{\pi(R)}})$ of $(\mathcal{V}_{T_1}, \ldots, \mathcal{V}_{T_R})$ such that:

$$\exists v \in [R]: \ s^{(t)} \in \mathcal{V}_{T_{\pi(v)}}, \ s^{(t+1)} \in \mathcal{V}_{T_{\pi(v+1)}}, \ \ldots, \ s^{(t+w-1)} \in \mathcal{V}_{T_{\pi(v+w-1)}}. \quad (3)$$

---

**Algorithm 1** Pattern-based Watermarking

---

**Input:** Base LLM $P_M$, a pre-specified pattern $\mathcal{P}$, the partition $\{\mathcal{V}_{T^{(1)}}, \ldots, \mathcal{V}_{T^{(r)}}\}$, and $\delta > 0$.
**Output:** Generated text $s^{(1:T)}$.
1: Initialize $t \leftarrow 1$, prompt $s^{(-N_p:0)}$.
2: **while** $t \leq T$ **do**
3:      Get current tag $T_{(t \bmod R)+1}$ from pattern at step $t$.
4:      Compute base logits $l_u^{(t)}$, $u \in \mathcal{V}$.
5:      Apply logit shift for $u \in \mathcal{V}_{T_{(t \bmod R)+1}}$ and sample $s^{(t)} \sim \tilde{p}^{(t)}$ according to equation 2.
6:      $t \leftarrow t + 1$.
7: **end while**
8: **return** $\{s^{(1:T)}\}$.

---

We denote
$$I_w(t) = \mathbb{1}\left\{\exists \text{ cyclic shift } \pi \text{ such that equation 3 is satisfied}\right\},$$
which is a binary indicator on whether the subsequence $s^{(t:t+w-1)}$ aligns with the watermark pattern.

**Watermark Detection Statistic.** Given the pre-specified pattern $\mathcal{P}$ and the window size $w$, we define the detection statistic $|\mathbf{s}|_D$ as the normalized count of matching subsequences:

$$|\mathbf{s}|_D = \frac{1}{T-w+1} \sum_{t=1}^{T-w+1} I_w(t). \tag{4}$$

The value of $|\mathbf{s}|_D$ is then compared to a predefined threshold $\tau_d$ (chosen by controlling false alarm rate); when $|\mathbf{s}|_D \geq \tau_d$, we reject $\mathcal{H}_0$ and conclude the text is likely watermarked (see Algorithm 2).

---

**Algorithm 2** Pattern-based Watermark Detection

---

**Input:** Text $s^{(1:T)}$, pattern $\mathcal{P}$, detection threshold $\tau_d$.
**Output:** Decision (watermarked or not).
1: Compute detection statistics $|\mathbf{s}|_D$ in equation 4.
2: **if** $|\mathbf{s}|_D \geq \tau_d$ **then**
3:      **return** Watermarked.
4: **else**
5:      **return** Not watermarked.
6: **end if**

---

---

**Algorithm 3** Edit Detection for Pattern-based Watermarking

---

**Input:** Text $s^{(1:T)}$, watermarking pattern, detection threshold $\tau_e$.
**Output:** Decision (edited or not) and the potential edit region.
1: Compute token-specific detection statistics $|\mathbf{s}|_E(t)$ as in equation 5 for all $t$.
2: **if** $\min_{t=w,\ldots,t-w+1} |\mathbf{s}|_E(t) < \tau_e$ **then**
3:      **return** Edit detected; and return the indexes set $I = \{t : |\mathbf{s}|_E(t) < \tau_e\}$.
4: **else**
5:      **return** Not edited.
6: **end if**

---

### 3.3 POST-GENERATION EDIT DETECTION

We then present our lightweight edit detection statistics designed to identify local positions that violate the pre-specified pattern; see a proof-of-concept illustration in Figure 2. We will again use the binary indicator $I_w(t)$ as the crucial element in constructing the edit detection statistics. We define the local edit statistic at each token index $t$ as, again, for a window of size $w$:

$$|\mathbf{s}|_E(t) = \frac{1}{w} \sum_{i=0}^{w-1} I_w(t-i). \tag{5}$$

Intuitively, the above average computes the average alignment of these $w$ windows, which all contain the current token $s^{(t)}$, with the pattern $\mathcal{P}$. We then compare each local statistic $|\mathbf{s}|_E(t)$ with a pre-specified threshold (calibrated to control the false alarm rate), and output all regions with statistics below the threshold. See Algorithm 3 for a complete summary of the procedure. Detailed computational complexity analysis is provided in Appendix C.

We present the following guarantee on the false alarm rate of edit detection under certain assumptions. The proof can be found in Appendix A.

**Theorem 3.1** (Type-I error rate of edit detection)**.** *Assume that under a clean watermark, the pattern alignment probability for each window of size $w$ is $\mu_1^{(w)} := \mathbb{P}[I_w(t) = 1], \forall t$. When $\mu_1^{(w)} = 1$ (hard watermarking with strict pattern adherence), we have the Type-I error rate (probability of a false alarm) $\Pr[|\mathbf{s}|_E(t) < \tau_e \mid \text{no edit}] = 0$ for any $\tau_e < 1$. When $\mu_1^{(w)} < 1$ (soft watermarking), we have for any detection threshold $\tau_e < \mu_1^{(w)}$, the Type-I error rate at token $t$ under a clean (unedited) watermark is bounded by*

$$\Pr[|\mathbf{s}|_E(t) < \tau_e \mid \text{no edit}] \leq \exp\left(-\frac{w^2(\mu_1^{(w)} - \tau_e)^2}{2\Delta^{(w)}}\right),$$

*where $\Delta^{(w)} := \sum_{i,j} \mathbb{E}[I_w(t-i)I_w(t-j)]$ is a constant that depends on $w$.*

It can be seen that the false alarm probability generally remains small when the detection threshold is relatively low compared to the pattern alignment probability $\mu_1^{(w)}$. Though the exact value of $\mu_1^{(w)}$ is generally intractable, its lower bound typically depends on the entropy of the LLMs next-token distribution and increases with the watermarking strength parameter $\delta$ (Kirchenbauer et al., 2023a). It is also worthwhile noting that the constant $\Delta^{(w)}$ here reflects the complex dependencies among sliding windows, which are difficult to characterize explicitly in large language models. Moreover, establishing detection accuracy under edits is more challenging, as the edit process itself is not well modeled by simple statistical assumptions. Further analysis is provided in Appendix A.

## 4 NUMERICAL EXPERIMENTS

**Experimental Setup.** We simulate texts using two large language models: LLaMA-2-7B and OPT-1.3b, both accessed via Hugging Face Transformers with deterministic decoding with 4-beam search. In all experiments, prompts are a sample of WikiText texts (Merity et al., 2017). The generated texts are all embedded with the combinatorial pattern-based watermarks with varying watermarking strength $\delta$. The edited texts are then generated by specifying three types of edits: token replacement, insertion, and deletion, and the length of each consecutive edit, ranging from 1 token to 6 tokens long. Random edits are injected in randomly selected contiguous spans. See Figure 1 for an overview of the simulated data we used in our numerical experiments.

**Evaluation.** We conduct two sets of evaluations. First, we evaluate the edit detection performance. For each edited text, we compute token-level edit detection statistics and compare them against a pre-selected threshold. The thresholds are calibrated to control the Type-I error rate (i.e., false alarm rate) at 0.1 across all experiments. We set the window size as $w = 8$ for the longer pattern in Example 3.2 and $w = 2$ for all other cases including the baselines. We report both illustrative examples of the edit detection statistics (in Figure 4) and the average detection accuracy across different edit types and lengths (in Figure 5). Second, we evaluate watermark detectability to ensure that the pattern-based watermark remains identifiable. We also illustrate the fundamental trade-off between detection effectiveness and the perplexity (i.e., text quality) of the watermarked outputs.

**Runtime Performance.** All experiments were conducted on an RTX 6000Ada GPU with 48GB of VRAM. The detection process is relatively efficient, taking less than one second to perform both watermark and edit detection on a batch of 64-token texts generated from 32 prompts. Table 1 shows the text generation time with and without applying the watermark. Note that the generation time is only incurred during dataset construction for evaluation purposes.

Table 1: Text Generation Wall Time: Watermarked vs. Unwatermarked. This measurement is done on RTX PRO 6000 WS using meta-llama/Llama-2-13b-hf.

| Tokens Generated | Wall Time (seconds) | |
| --- | --- | --- |
| | No Watermark | Watermarked (AB) |
| 32 (rows) x 64 (tokens) = 2,048 | 4.22 | 4.33 |
| 32 (rows) x 512 (tokens) = 16,384 | 86.65 | 86.63 |

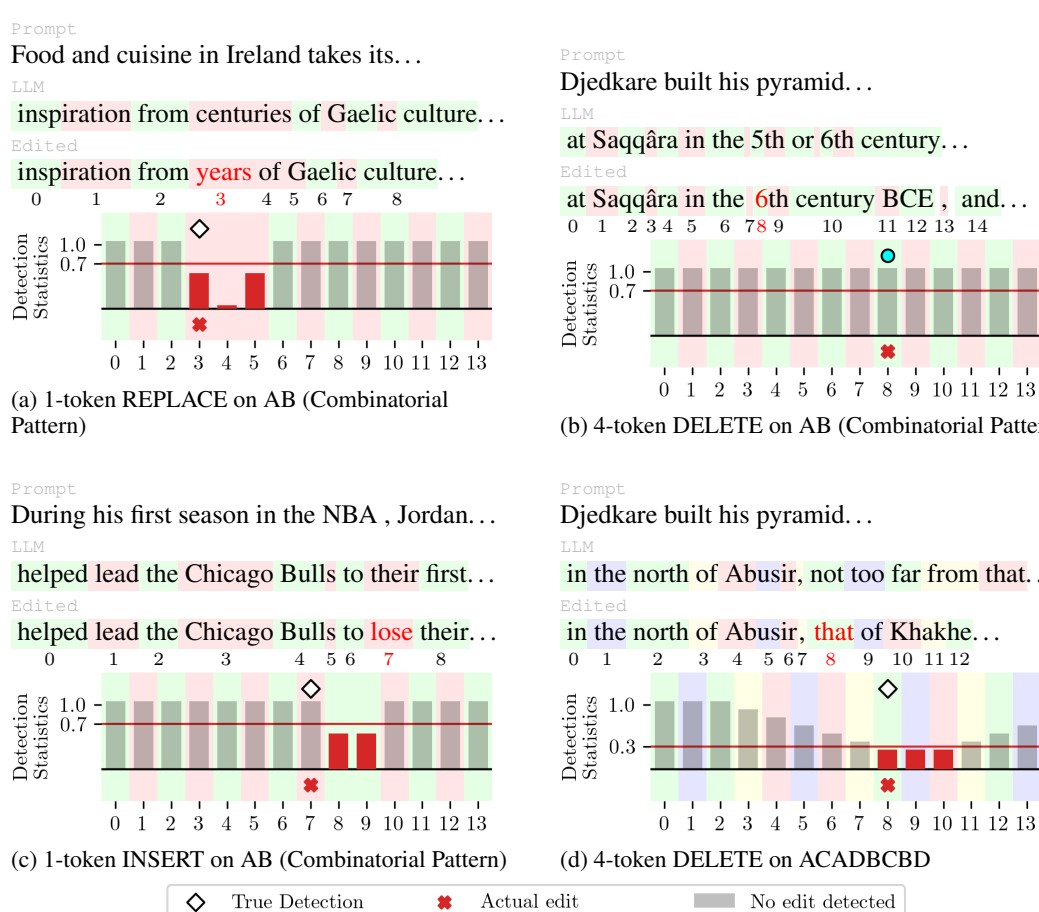

(a) 1-token REPLACE on AB (Combinatorial Pattern)

(b) 4-token DELETE on AB (Combinatorial Pattern)

(c) 1-token INSERT on AB (Combinatorial Pattern)

(d) 4-token DELETE on ACADBCBD

Figure 4: Four examples of edit detection statistics under the two combinatorial patterns. Each example shows the prompt text, the watermarked LLM-generated text, and the edited text. The detection threshold is marked in red, and detected edit spans are represented by red bars that fall below the threshold. We mark the true detection and missed detection in the plot, under a tolerance of $L = 3$. The examples are generated using LLaMA-2-7b with watermarking strength $\delta = 5.8$.

## 4.1 RESULTS ON POST-GENERATION EDIT DETECTION

We first evaluate the performance of our detection method across three canonical types of post-generation edits: replacement, insertion, and deletion. We demonstrate that the pattern-based watermark allows accurate localization of the edited spans. For each type of edit, Figure 4 shows the edit detection statistics across token positions. It can be seen that the edit detection statistics fall below the threshold in almost all edited tokens, indicating efficient detection of local edits. In contrast, the edit detection statistics lie above the threshold during most non-edit regions, as the threshold is calibrated to achieve a small Type-I error rate. Furthermore, comparing Figure 4 (b) and (d), we observe that the longer combinatorial pattern in (d) is more effective at detecting certain edit lengths, particularly in the case of deletions.

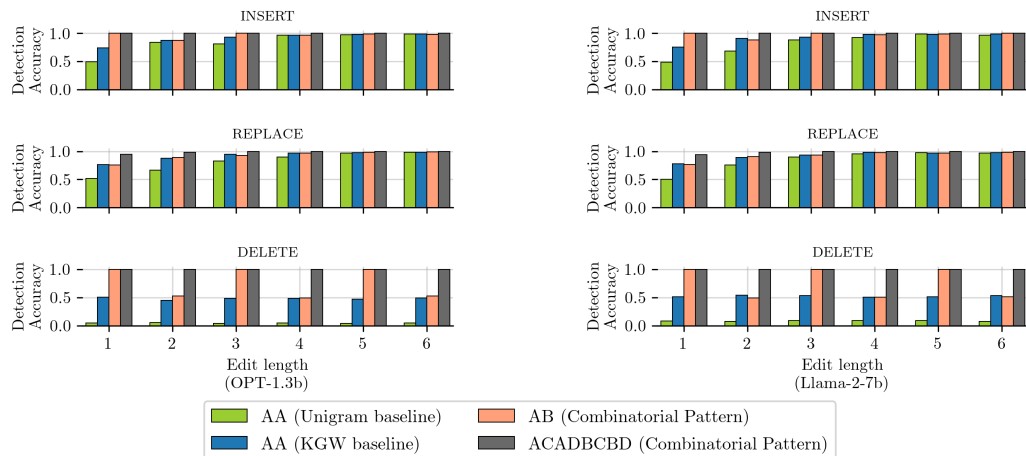

Figure 5: Edit detection accuracy under different edit lengths (1 to 6 tokens) and three edit types (insertion, replacement, and deletion) on OPT-1.3b (left) and Llama-2-7b (right). The watermarking strength parameter is $\delta = 5.8$. The exact numerical values for edit detection accuracy can be found in Table 2 in the Appendix.

Furthermore, we compute the average edit detection accuracy over 1,000 samples of 64 tokens long, using a fixed Type-I error rate of 0.1 and a tolerance parameter $L = 3$. Results for both LLaMA-2-7b and OPT-1.3b are shown in Figure 5, evaluated under various combinatorial patterns with a fixed watermark strength. We also compare against baseline watermarking methods, including KGW (Kirchenbauer et al., 2023a) and Unigram (Zhao et al., 2024a), both perturb logits over a selected green list. For a fair comparison, we adapt our local edit detection statistic to these settings by treating them as having a degenerate pattern of the form $AA \cdots$, where $A$ refers to the green list.

As shown in Figure 5, the proposed method can detect various post-generation edits with high accuracy, especially when using the longer combinatorial pattern. The detection accuracy generally increases quickly with the edit span, indicating that consecutive edits are easier to detect. Combinatorial patterns significantly outperform baseline KGW and Unigram watermarking methods, especially for detecting deletion-type edits and short-span edits (a particularly challenging case). This is very promising given the simplicity of the proposed edit detection scheme. For a simple pattern with a period of two (the $AB$ pattern used here), it is hard to detect deletions that align exactly with the pattern (e.g., removals of length two, four, and six in Figure 5). However, the longer combinatorial pattern can achieve high detection accuracy for such token deletion edits. See Appendix B.2 for additional results under varying watermarking strengths, patterns, and sampling mechanisms.

## 4.2 RESULTS ON WATERMARK DETECTION

To evaluate watermark detectability, we generate both unwatermarked and watermarked texts of 64 tokens long, on LLaMA-2-7b and OPT-1.3b. We then apply the watermark detection statistics equation 4 to distinguish between watermarked and unwatermarked texts. The detection threshold is selected to control the Type-I error rate at 0.1, and we report the corresponding Type-II error rate to assess detection effectiveness. Meanwhile, to assess the impact of watermarking on text quality, we compute the perplexity (PPL) of the generated text.

In Figure 6, we plot both the Type-II error rate and PPL across varying watermarking strength $\delta$ and different combinatorial patterns. We also include the perplexity of the unwatermarked model for comparison. As the watermark strength $\delta$ increases, we observe a general decrease in the Type-II error rate and an increase in PPL, indicating that the combinatorial watermark becomes more detectable but the generated text is of lower quality. This highlights the fundamental trade-off between watermark detectability and generation quality. Furthermore, the longer combinatorial pattern—with four unique tags $(A, B, C, D)$—exhibits the weakest trade-off between detectability and text quality. A possible explanation is that increasing the number of unique tags reduces the size of each sub-vocabulary. This, in turn, degrades text quality and thus increases PPL.

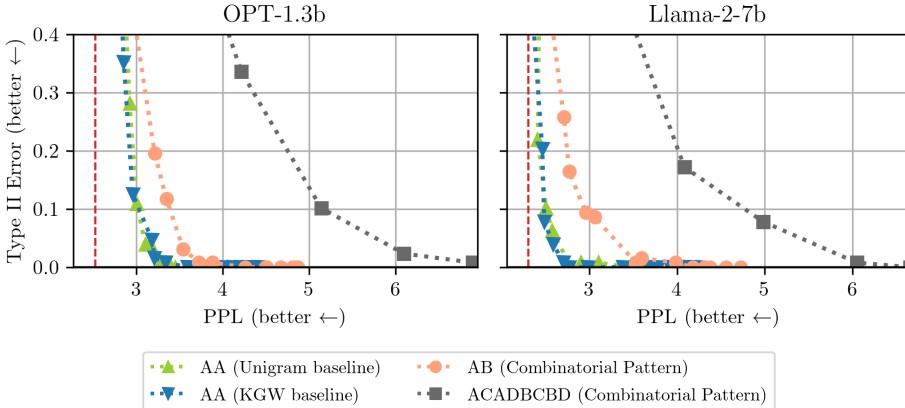

Figure 6: Tradeoff curve between the Type-II error rate of watermark detection and the perplexity (PPL) of generated text. The red dashed line indicates the perplexity of unwatermarked text.

## 5 CONCLUDING REMARKS

We formulate and study the problem of local edit detection in watermarked LLM outputs. We introduce a combinatorial pattern-based watermark that embeds rich local structure into the watermarked text. Leveraging this structure, we derived lightweight statistics that can flag and localize suspect spans containing edits. We evaluate the edit detection performance via experiments across various editing scenarios. There are still several limitations of our work. For example, the pattern design space explored is relatively narrow with at most four unique tags, and the method remains less effective for very short edits (one or two tokens), which are challenging to detect. Moreover, we focus on lightweight detection statistics such as equation 4, which makes minimal assumptions about the underlying pattern. However, the trade-off between watermark detectability, edit detection accuracy, and perplexity could potentially be improved by adopting a more sophisticated detection method. To tackle these challenges, future work includes exploring longer or adaptive pattern designs, further improving the detection accuracy, and extending edit detection to other watermarking frameworks.

ETHICS STATEMENT

This work adheres to the ICLR Code of Ethics. Our research does not involve human subjects, sensitive personal data, or experiments that may pose harm to individuals or communities.

REPRODUCIBILITY STATEMENT

All theoretical assumptions and complete proofs of the main results are provided in the appendix A. Detailed descriptions of the experimental settings, including data generation, preprocessing, parameter choices, and evaluation metrics, can be found in Section 4, and the source code can be found in the supplementary materials.

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

## A PROOFS FOR SECTION 3 AND MORE THEORETICAL ANALYSIS

*Proof to Theorem 3.1.* Recall that the edit detection statistic

$$|\mathbf{s}|_E(t) = \frac{1}{w} \sum_{i=0}^{w-1} I_w(t-i),$$

is the normalized count of sliding windows of length $w$ that perfectly match the known tag pattern $\mathcal{P}$. Here each indicator $I_w(t-i)$ takes value 1 if and only if

$$(s^{(t-i)}, \ldots, s^{(t-i+w-1)}) \in \mathcal{V}_{T_1} \times \cdots \times \mathcal{V}_{T_w},$$

i.e., the generated tokens in the window fall entirely in the corresponding subset prescribed by $\mathcal{P}$. Here, with a slight abuse of notation but for simplicity, we just use $\{\mathcal{V}_{T_1} \cdots \mathcal{V}_{T_w}\}$ to denote the pattern enforced to tokens within the current sliding window.

Under a clean watermark, by our assumption and the construction of the watermarking scheme, for each window indicator we have $\mathbb{E}[I_w(t-i)] = \mu_1^{(w)}$ and hence $\mathbb{E}[|\mathbf{s}|_E(t)] = \mu_1^{(w)}$.

We consider two cases separately. First, if $\mu_1^{(w)} = 1$, i.e., the so-called hard watermarking regime (Kirchenbauer et al., 2023a) where the token is strictly required to be drawn from the corresponding list. For example, this can happen when we set the watermarking strength parameter $\delta$ to be large. Under this case, we have $|\mathbf{s}|_E(t) \equiv 1$ and thus $\Pr[|\mathbf{s}|_E(t) < \tau_e \mid \text{no edit}] = 0$, which implies no false alarm.

For the soft watermarking regime with $\mu_1^{(w)} < 1$, in such cases, the token is more likely to be drawn from the corresponding list but is not guaranteed. Note that the list of indicators $\{I_w(t-i)\}_{i=0}^{w-1}$ is not independent, thus the indicators can be represented by a dependency graph: two indicators are adjacent if their corresponding windows overlap. For windows with offset $|i - j| = k < w$, the overlap size is $w - k$, we have the joint probability is

$$\mathbb{E}[I_w(t-i)I_w(t-j)] = \mu_1^{(w+k)}.$$

Define the $w$-dependent constant $\Delta^{(w)} := \sum_{i,j} \mathbb{E}[I_w(t-i)I_w(t-j)] = w\mu_1^{(w)} + 2\sum_{k=1}^{w-1}(w - k)\mu_1^{(w+k)}$. Janson's inequality (Janson, 1990) states that for $z < \mathbb{E}[\sum_{i=0}^{w-1} I_w(t-i)] = w\mu_1^{(w)}$,

$$\mathbb{P}\left(\sum_{i=0}^{w-1} I_w(t-i) < z\right) \leq \exp\left(-\frac{(\mathbb{E}[\sum_{i=0}^{w-1} I_w(t-i)] - z)^2}{2\Delta^{(w)}}\right).$$

For the false alarm event $\{|\mathbf{s}|_E(t) < \tau_e\}$ we have $\sum_{i=0}^{w-1} I_w(t-i) < w\tau_e$, and thus we have

$$\mathbb{P}[|\mathbf{s}|_E(t) < \tau_e \mid \text{no edit}] \leq \exp\left(-\frac{w^2(\mu_1^{(w)} - \tau_e)^2}{2\Delta^{(w)}}\right).$$

$\square$

**Discussion.** In practice, the pattern alignment probability $\mu_1^{(w)}$ is generally tractable and there is no closed-form expression. However, the probability for token-level adherence is given in Lemma E.1 in Kirchenbauer et al. (2023a). For example, under watermarking parameter $\delta$, assuming the two sub-vocabulary sets are of equal size, the probability of drawing a token from the current target subset is lower bounded by $\tilde{\mu}_1 := \frac{\frac{1}{2}\alpha}{1 + \frac{1}{2}(\alpha-1)} S\left(p, \frac{\frac{1}{2}(\alpha-1)}{1 + \frac{1}{2}(\alpha-1)}\right)$, where $\alpha = e^\delta$ and $S(p, z) := \sum_k \frac{p_k}{1 + zp_k}$, where $p$ represent the next-token probability (Kirchenbauer et al., 2023a).

It should be noted that the $w$-dependent constant $\Delta^{(w)}$ can scale as $O(w^2)$ in the worst case under strong positive correlations among overlapping windows, leading to a relatively loose upper bound on the false alarm rate. On the other hand, if overlapping windows are independent—so that pattern alignment in one window does not affect another—Hoeffdings inequality yields a much tighter

bound that can decay exponentially with $w$. In practice, however, the exact dependence structure among sliding windows in large language models is intricate and difficult to characterize.

We also note that the analysis on the edit detection accuracy would be much more complicated due to the complication of all possible edits. As an example, in the following, we provide an analysis by assuming that the edit happens in such a way that reduces the pattern alignment probability for each sliding window to $\mu_0^{(w)} := \mathbb{P}[I_w(t) = 1], \forall t$ and $\mu_0^{(w)}$ is much smaller than the pattern alignment probability $\mu_1^{(w)}$ when there is no edits (clean watermark). We can apply Theorem 5 in Janson & Ruciński (2002) to obtain an upper bound for the miss detection probability. Specifically, by Theorem 5 in Janson & Ruciński (2002) and under our assumption, we have

$$\mathbb{P}[|\mathbf{s}|_E(t) \geq \tau_e \mid \text{edit}] \leq w \exp\left(-\frac{w^2(\tau_e - \mu_0^{(w)})^2}{4w(w\mu_0^{(w)} - w(\tau_e - \mu_0^{(w)})/3)}\right) = w \exp\left(-\frac{3(\tau_e - \mu_0^{(w)})^2}{4(2\mu_0^{(w)} + \tau_e)}\right).$$

It should be mentioned that the upper bound may not be very informative due to two reasons. First, it is generally much larger than the lower tail probability, as capturing upper-tail probabilities with overlapping time windows is intrinsically difficult (Janson & Ruciński, 2002). Second, here we are assuming that the edits reduce the pattern alignment probability uniformly for each window to $\mu_0^{(w)}$, which could be unrealistic in practice. For example, in many cases, a local token insertion may force $\mu_0^{(w)} = 0$ for windows containing the edit, resulting in 100% detection accuracy. Therefore, we rely primarily on the empirical results in Section 4 to demonstrate the effectiveness of edit detection across scenarios.

**Analysis on watermark detectability.** We also provide a watermark detectability error analysis for completeness and to support the design of our combinatorial watermarking. The results show that the detection accuracy converges to 1 as $T \to \infty$, ensuring reliable detection with sufficiently long watermarked text. Likewise, the Type-I error rate converges to 0 as $T \to \infty$, implying that sufficiently long unwatermarked text yields a vanishing false alarm rate.

**Theorem A.1** (Watermark detection error rates). *Assume that under a clean watermark, the pattern alignment probability for each window of size $w$ is $\mu_1^{(w)} := \mathbb{P}[I_w(t) = 1], \forall t$. When there is no watermarking, assume this probability is reduced to $\mu_0^{(w)} < \mu_1^{(w)}$. Assume the observed data contains $T$ tokens in total, and the window size is $w$ in the detection statistics.*

- *The probability of detecting the watermark, for a given detection threshold $\tau_D$, is at least*

$$\mathbb{P}(|\mathbf{s}|_D \geq \tau_d) \geq 1 - \exp\left(-(T - w + 1)\frac{(\mu_1^{(w)} - \tau_d)^2}{2w\mu_1^{(w)}}\right).$$

- *The Type-I error rate (probability of false alarm) when there is no watermarking is*

$$\mathbb{P}[|\mathbf{s}|_D \geq \tau_d] \leq w \cdot \exp\left(-(T - w + 1)\frac{3(\tau_d - \mu_0^{(w)})^2}{4w \cdot (2\mu_0^{(w)} + \tau_d)}\right).$$

*Proof.* Recall that the global watermark detection statistic

$$|\mathbf{s}|_D = \frac{1}{T - w + 1} \sum_{t=1}^{T-w+1} I_w(t),$$

is the normalized count of sliding windows of length $w$ that perfectly match the known tag pattern.

Under the watermarking regime, similar to the proof of Theorem 3.1, by our assumption and the construction of the watermarking scheme, for each window indicator we have $\mathbb{E}[I_w(t - i)] = \mu_1^{(w)}$ and hence $\mathbb{E}[|\mathbf{s}|_D(t)] = \mu_1^{(w)}$. Again we define the $(T, w)$-dependent constant

$$\Delta^{(T,w)} := \sum_{\substack{(i,j): \\ |i-j|<w}} \mathbb{E}[I_w(i)I_w(j)] \leq (T-w+1)\mu_1^{(w)} + (T-w+1)(w-1)\mu_1^{(w)} = (T-w+1)w\mu_1^{(w)}.$$

Then we can apply Jansons inequality (Janson, 1990), which guarantees for $\tau_d < \mu_1^{(w)}$,

$$\mathbb{P}(|\mathbf{s}|_D \leq \tau_d) = \mathbb{P}(\sum_{t=1}^{T-w+1} I_w(t) < (T-w+1)\tau_d) \leq \exp\left(-(T-w+1)^2 \frac{(\mu_1^{(w)} - \tau_d)^2}{2\Delta^{(T,w)}}\right).$$

By substituting the upper bound to $\Delta^{(T,w)}$, we can further simplify the above inequality as

$$\mathbb{P}(|\mathbf{s}|_D \leq \tau_d) \leq \exp\left(-(T-w+1)\frac{(\mu_1^{(w)} - \tau_d)^2}{2w\mu_1^{(w)}}\right).$$

For the false alarm event $\{|\mathbf{s}|_D \geq \tau_d\}$ when there is no watermarking, we have

$$\mathbb{P}[|\mathbf{s}|_D \geq \tau_d] \leq w \cdot \exp\left(-(T-w+1)^2 \frac{(\tau_d - \mu_0^{(w)})^2}{4w \cdot ((T-w+1)\mu_0^{(w)} + (T-w+1)(\tau_d - \mu_0^{(w)})/3)}\right)$$

$$\leq w \cdot \exp\left(-(T-w+1)\frac{3(\tau_d - \mu_0^{(w)})^2}{4w \cdot (2\mu_0^{(w)} + \tau_d)}\right).$$

$\square$

# B   ADDITIONAL ALGORITHMIC DETAILS AND EXPERIMENTAL RESULTS

## B.1   CONCRETE EXAMPLES OF DETECTION STATISTICS

To better illustrate the detection algorithms, we give concrete examples of the constructed watermark detection and edit detection statistics for the two exemplary combinatorial patterns in Example 3.1 and Example 3.2.

**Example 3.1.** With two unique tags (e.g., $A$ and $B$), we define the pattern $\mathcal{P} = \{A, B\}$, and thus the watermark is governed by the order $A, B, A, B, \ldots$. Here $A$ and $B$ can be interpreted as the green and red lists (see Figure 2), respectively, aligning with standard terminology in prior work. In the following, with a slight abuse of notation, we use $A$ and $B$ to also denote their corresponding subset of vocabulary, when not causing confusion.

Based on the definition in equation 3, we have the following concrete formulations for $I_w(t)$, which is the core component in our watermark detection and edit detection statistics.

- For window size $w = 2$, we have
$$I_w(t) = \mathbb{1}\left\{s^{(t)} \text{ and } s^{(t+1)} \text{ belongs to different sets } (A, B \text{ or } B, A)\right\}.$$

- For window size $w = 4$, we have
$$I_w(t) = \mathbb{1}\left\{s^{(t)}, s^{(t+1)}, s^{(t+2)}, s^{(t+3)} \text{ are in sets } A, B, A, B \text{ or } B, A, B, A\right\}.$$

This also illustrates the case when the window size $w$ exceeds the pattern period.

**Example 3.2.** With four unique tags (e.g., $A, B, C, D$), we define the combinatorial pattern $\mathcal{P} = \{A, C, A, D, B, C, B, D\}$, and the watermark is governed by the order $A, C, A, D, B, C, B, D, A, C, A, D, B, C, B, D, \ldots$.

Similarly, we have the following concrete formulation of $I_w(t)$ that can be efficiently computed for performing watermark detection and edit detection:

- For window size $w = 2$, we have
$$I_w(t) = \mathbb{1}\left\{s^{(t)} \text{ and } s^{(t+1)} \text{ are in } AC, CA, AD, DB, BC, CB, BD, \text{ or } DA\right\}.$$

- For window size $w = 4$, we have
$$I_w(t) = \mathbb{1}\{s^{(t:t+3)} \text{ are in } ACAD, CADB, ADBC, DBCB,$$
$$BCBD, CBDA, BDAC, \text{ or } DACA\}.$$

The statistics for larger window sizes are similarly constructed based on the same principle.

**Insights for combinatorial pattern design.** Motivated by the detection statistics for watermarking, we list some insights for designing the combinatorial pattern to enable simple watermark detection. For ease of watermark detection, we may impose the following structural restriction on our patterns: we let both pattern length $R$ and number of unique tags $r$ to be even numbers, and the tag assignment alternates between even and odd indices—meaning $T_i = T^{(j)}$ only if $i$ and $j$ are both even or both odd. Both examples 3.1 and 3.2 satisfy such a property. This design enables simple and effective detection using the smallest window size $w = 2$. Specifically, we define the window indicator for positions $t$ as $I_w(t) = \mathbb{1}\left\{s^{(t)} \in \mathcal{V}_{odd} \text{ and } s^{(t+1)} \in \mathcal{V}_{even}, \text{ or } s^{(t)} \in \mathcal{V}_{even} \text{ and } s^{(t+1)} \in \mathcal{V}_{odd}\right\}$, where $\mathcal{V}_{odd} = \cup_{1 \le i \le r, i \text{ is odd}} \mathcal{V}_{T^{(i)}}$ and $\mathcal{V}_{even} = \cup_{1 \le i \le r, i \text{ is even}} \mathcal{V}_{T^{(i)}}$. In other words, detection could rely only on whether the observed token sequence follows the expected even-odd alternation. This simple test does not rely on the full pattern structure. In contrast, the full pattern sequence provides richer information that can be leveraged to localize specific edit positions.

*Remark* B.1. We also note that the above watermark detection is performed for black-box LLMs. In practice, when we do have access to the logits information (such as in white-box LLMs), we can instead use the log-likelihood ratio as our edit detection statistics and watermark detection statistics, which will yield more accurate detection results due to the utilization of more information. And this can potentially improve both the edit detection accuracy and the watermark detection accuracy.

B.2   MORE NUMERICAL RESULTS

**Supplementary Results to Figure 5.** To enable precise comparison across different methods and edit conditions, we provide the table below, which reports the exact numerical values for edit detection accuracy as shown in Figure 5.

Table 2: Edit detection accuracy under different edit lengths (1 to 6 tokens) and three edit types (insertion, replacement, and deletion) on OPT-1.3b (left) and Llama-2-7b (right). The watermarking strength parameter is $\delta = 5.8$. In all cases we allow an evaluation tolerance of $L = 3$ tokens. This corresponds to Figure 5.

| Edit Type/Count | | OPT-1.3b | | | | Llama-2-7b | | | |
| --- | --- | --- | --- | --- | --- | --- | --- | --- | --- |
| | | AA (Unigram) | AA (KGW) | AB | ACAD BCBD | AA (Unigram) | AA (KGW) | AB | ACAD BCBD |
| INSERT | 1 | 0.51 | 0.75 | 1.00 | 1.00 | 0.51 | 0.66 | 1.00 | 1.00 |
| | 2 | 0.78 | 0.85 | 0.89 | 1.00 | 0.78 | 0.89 | 0.84 | 1.00 |
| | 3 | 0.88 | 0.94 | 0.98 | 1.00 | 0.86 | 0.93 | 0.97 | 1.00 |
| | 4 | 0.95 | 0.95 | 0.96 | 1.00 | 0.93 | 0.96 | 0.96 | 1.00 |
| | 5 | 0.96 | 0.98 | 0.96 | 1.00 | 0.96 | 0.96 | 0.98 | 1.00 |
| | 6 | 0.97 | 0.96 | 0.98 | 1.00 | 0.97 | 0.98 | 0.97 | 1.00 |
| REPLACE | 1 | 0.49 | 0.71 | 0.71 | 0.97 | 0.46 | 0.80 | 0.78 | 0.94 |
| | 2 | 0.76 | 0.89 | 0.87 | 0.99 | 0.79 | 0.80 | 0.90 | 0.99 |
| | 3 | 0.88 | 0.95 | 0.93 | 1.00 | 0.89 | 0.93 | 0.96 | 1.00 |
| | 4 | 0.92 | 0.96 | 0.96 | 0.99 | 0.96 | 0.98 | 0.95 | 0.99 |
| | 5 | 0.96 | 0.97 | 0.96 | 1.00 | 0.94 | 0.97 | 0.96 | 1.00 |
| | 6 | 0.96 | 0.98 | 0.98 | 1.00 | 0.96 | 0.98 | 0.98 | 1.00 |
| DELETE | 1 | 0.01 | 0.45 | 1.00 | 1.00 | 0.06 | 0.55 | 1.00 | 1.00 |
| | 2 | 0.01 | 0.48 | 0.48 | 1.00 | 0.04 | 0.45 | 0.49 | 1.00 |
| | 3 | 0.03 | 0.45 | 1.00 | 1.00 | 0.05 | 0.48 | 1.00 | 1.00 |
| | 4 | 0.00 | 0.45 | 0.50 | 1.00 | 0.04 | 0.50 | 0.50 | 1.00 |
| | 5 | 0.02 | 0.46 | 1.00 | 1.00 | 0.05 | 0.56 | 1.00 | 1.00 |
| | 6 | 0.01 | 0.42 | 0.51 | 1.00 | 0.04 | 0.47 | 0.50 | 1.00 |

**Results With Varying Watermarking Strength $\delta$, Combinatorial Pattern, and Multinomial Sampling.** We present more results on the average edit detection accuracy under a variety of wa-

termarking strengths in Figure 7. We also included a new pattern with $r = 3$ unique tags. We note that higher watermarking strength increases accuracy in general. We also note that for the longer ACADBCBD combinatorial pattern, it becomes more effective only beyond a certain watermarking strength threshold. This is likely because, under lower watermarking strengths, the generated watermarked text does not reliably adhere to the pattern, thus the edit detection is less effective. As the watermarking strength increases, the watermarked text has better adherence to the pattern, leading to better edit detection performance.

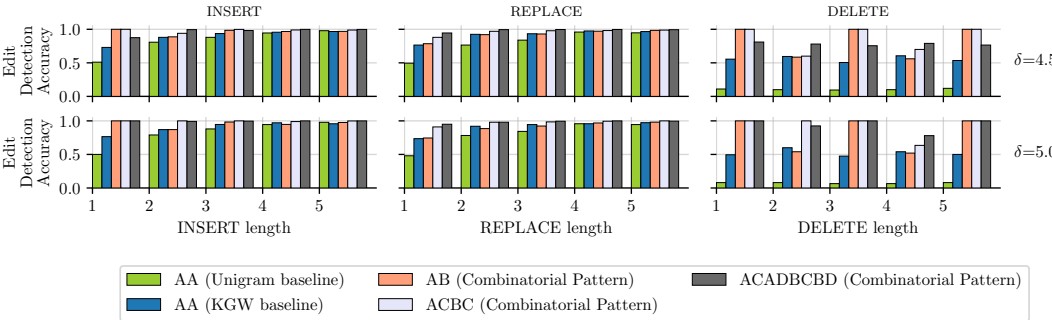

Figure 7: Detection accuracy vs edit type and length under different watermarking strengths $\delta$. This was generated in a similar fashion to Figure 5 using Llama-2-7b, with 200 samples of 100-token long generated text for each combination of edit type, length, watermarking strength, and pattern. The texts in this case are generated using multinomial sampling where (`temperature=1.0`, `top_p=0.8`).

We report edit detection accuracies and false positive rates (FPRs) under $\delta \in \{2.5, 3.5, 4.5, 5.5\}$ for the baselines Unigram, KGW, and three proposed patterns (Table 3). Each accuracy is computed on 256 samples of 128-token text generated via multinomial sampling (temperature = 1.0, top_p = 0.8), followed by contiguous edits of varying widths. Edit detection uses a $\pm 2$ token tolerance. Results show that the AB pattern remains effective at lower $\delta$ values with a 10% FPR. It is worthwhile mentioning that while the baseline KGW method appears to have higher detection accuracy, it is at the cost of a higher FPR. This reflects the inherent tradeoff between detection accuracy and false positives, which is directly influenced by the choice of $\delta$. Intuitively, larger $\delta$ values can ensure a small FPR and preferable detection accuracies, whereas smaller $\delta$ increases text perplexity and leads to a less favorable balance between accuracy and FPR.

Furthermore, to illustrate that our watermarking method is compatible with random vocabulary partitions at each time step, we also include a randomized variant of the AB pattern, denoted "AB (rand)" in Table 3. This variant randomizes the green/red vocabulary split at each token index $t$ based on the previous token $s^{(t-1)}$, following a mechanism similar to KGW (Kirchenbauer et al., 2023a). Specifically, we compute a hash of the previous token, use it to seed a random number generator, and then partition the vocabulary into green and red lists according to that seed. It can be seen from Table 3 that introducing such randomized vocabulary partitions improves edit detection accuracy. One possible reason is that after edits, the green/red partition used during generation becomes misaligned with the partition reconstructed via the hash function; this may cause more pattern violations as compared with a fixed vocabulary partition. It is also worth noting that under such randomized vocabulary partitions, our AB-pattern watermark attains substantially better edit-detection accuracy than the KGW baseline for $\delta \geq 3.5$. Although AB (rand) still underperforms KGW at $\delta = 2.5$, the performance gap is much smaller than that between the non-randomized AB pattern and KGW. Finally, our method continues to achieve a controlled false positive rate (FPR) of 10%, whereas KGWs slightly higher edit-detection accuracy comes at the cost of a larger FPR, which is less desirable in high-stakes applications.

We also report perplexity (PPL) as a measure of text quality in Table 3. It can be seen that our proposed watermarking method under the simplest AB pattern achieves perplexity comparable to the baseline methods Unigram and KGW across various $\delta$ values. More complex patterns may yield higher perplexity, which reflects a potential decrease in text quality, but we emphasize that this effect arises from a fundamental trade-off between edit detectability and output quality: longer patterns strengthen edit detection but may modestly reduce text quality. This trade-off is fully tunable through

the choice of pattern structure and watermarking strength, allowing practitioners to balance detection and quality according to specific requirements. Generally speaking, smaller watermarking strength $\delta$ and shorter patterns are sufficient to achieve a good balance between text quality and edit-detection performance; higher watermarking strength and longer patterns can be used when near-perfect edit detection is required, but may come with a slight reduction in text quality. Importantly, the impact on text quality is modest in our experiments. To provide qualitative evidence, we have included generated text examples under two prompts and multiple $\delta$ values in Tables 7 and 8 in Appendix D.

Table 3: Detection accuracy across edit types and lengths for Llama-2-7b under varying watermark strengths $\delta$. We pick the edit detection threshold such that FPR is below 0.1; when this is not attainable at a given $\delta$, we use the threshold chosen from the nearest higher $\delta$.

| Pattern | $\delta$ | PPL ↓ (base=4.93) | FPR ↓ | DELETE ↑ | | | | INSERT ↑ | | | | REPLACE ↑ | | | |
|---|---|---|---|---|---|---|---|---|---|---|---|---|---|---|---|
| | | | | 1 | 2 | 3 | 4 | 1 | 2 | 3 | 4 | 1 | 2 | 3 | 4 |
| Unigram | 2.50 | 6.97 | 0.15 | 0.53 | 0.53 | 0.53 | 0.56 | 0.75 | 0.86 | 0.93 | 0.92 | 0.72 | 0.84 | 0.92 | 0.96 |
| Unigram | 3.50 | 9.36 | 0.15 | 0.33 | 0.33 | 0.35 | 0.34 | 0.67 | 0.83 | 0.91 | 0.92 | 0.65 | 0.80 | 0.90 | 0.95 |
| Unigram | 4.50 | 13.31 | 0.10 | 0.17 | 0.18 | 0.18 | 0.18 | 0.59 | 0.79 | 0.90 | 0.91 | 0.55 | 0.76 | 0.88 | 0.95 |
| Unigram | 5.50 | 19.78 | 0.05 | 0.07 | 0.08 | 0.09 | 0.08 | 0.54 | 0.77 | 0.89 | 0.90 | 0.50 | 0.74 | 0.87 | 0.94 |
| KGW | 2.50 | 7.00 | 0.15 | 0.73 | 0.73 | 0.73 | 0.73 | 0.84 | 0.92 | 0.96 | 0.97 | 0.87 | 0.93 | 0.95 | 0.97 |
| KGW | 3.50 | 9.38 | 0.15 | 0.67 | 0.64 | 0.62 | 0.64 | 0.83 | 0.89 | 0.95 | 0.97 | 0.82 | 0.90 | 0.96 | 0.98 |
| KGW | 4.50 | 13.29 | 0.10 | 0.58 | 0.57 | 0.57 | 0.54 | 0.78 | 0.89 | 0.95 | 0.98 | 0.80 | 0.90 | 0.95 | 0.97 |
| KGW | 5.50 | 19.61 | 0.05 | 0.54 | 0.54 | 0.53 | 0.52 | 0.78 | 0.87 | 0.94 | 0.97 | 0.78 | 0.89 | 0.94 | 0.98 |
| AB | 2.50 | 6.96 | 0.10 | 0.38 | 0.65 | 0.36 | 0.63 | 0.51 | 0.64 | 0.69 | 0.67 | 0.62 | 0.61 | 0.65 | 0.69 |
| AB | 3.50 | 9.25 | 0.10 | 1.00 | 0.62 | 1.00 | 0.63 | 1.00 | 0.89 | 1.00 | 0.97 | 0.81 | 0.91 | 0.94 | 0.97 |
| AB | 4.50 | 12.97 | 0.10 | 1.00 | 0.55 | 1.00 | 0.59 | 1.00 | 0.90 | 1.00 | 0.97 | 0.78 | 0.89 | 0.94 | 0.97 |
| AB | 5.50 | 18.89 | 0.05 | 1.00 | 0.50 | 1.00 | 0.52 | 1.00 | 0.88 | 1.00 | 0.97 | 0.75 | 0.88 | 0.93 | 0.96 |
| AB (rand) | 2.50 | 6.96 | 0.10 | 0.62 | 0.64 | 0.64 | 0.64 | 0.69 | 0.65 | 0.73 | 0.70 | 0.76 | 0.74 | 0.74 | 0.74 |
| AB (rand) | 3.50 | 9.27 | 0.10 | 1.00 | 0.90 | 1.00 | 0.88 | 1.00 | 0.99 | 0.99 | 0.99 | 0.92 | 0.96 | 0.98 | 0.99 |
| AB (rand) | 4.50 | 13.00 | 0.05 | 1.00 | 0.85 | 1.00 | 0.84 | 1.00 | 0.97 | 0.99 | 0.99 | 0.88 | 0.97 | 0.99 | 0.99 |
| AB (rand) | 5.50 | 18.96 | 0.05 | 1.00 | 0.87 | 1.00 | 0.83 | 1.00 | 0.98 | 0.99 | 0.99 | 0.88 | 0.95 | 0.98 | 0.99 |
| ACBC | 2.50 | 7.06 | 0.35 | 0.82 | 0.84 | 0.84 | 0.88 | 0.89 | 0.97 | 0.96 | 0.98 | 0.94 | 0.98 | 0.98 | 0.98 |
| ACBC | 3.50 | 9.83 | 0.20 | 0.64 | 0.73 | 0.63 | 0.79 | 0.80 | 0.93 | 0.94 | 0.96 | 0.92 | 0.96 | 0.96 | 0.98 |
| ACBC | 4.50 | 14.77 | 0.20 | 1.00 | 1.00 | 1.00 | 0.75 | 1.00 | 1.00 | 1.00 | 1.00 | 0.92 | 0.98 | 0.99 | 1.00 |
| ACBC | 5.50 | 23.55 | 0.10 | 1.00 | 1.00 | 1.00 | 0.70 | 1.00 | 1.00 | 1.00 | 1.00 | 0.91 | 0.97 | 1.00 | 1.00 |
| ACADBCBD | 2.50 | 6.91 | 0.75 | 1.00 | 1.00 | 1.00 | 0.99 | 1.00 | 1.00 | 1.00 | 1.00 | 0.99 | 1.00 | 1.00 | 1.00 |
| ACADBCBD | 3.50 | 9.64 | 0.45 | 1.00 | 0.98 | 1.00 | 0.95 | 1.00 | 0.99 | 1.00 | 1.00 | 0.97 | 1.00 | 1.00 | 1.00 |
| ACADBCBD | 4.50 | 14.77 | 0.25 | 1.00 | 0.91 | 1.00 | 0.84 | 1.00 | 0.99 | 1.00 | 1.00 | 0.96 | 0.99 | 1.00 | 1.00 |
| ACADBCBD | 5.50 | 24.34 | 0.10 | 1.00 | 0.87 | 1.00 | 0.72 | 1.00 | 0.98 | 1.00 | 1.00 | 0.95 | 0.98 | 1.00 | 1.00 |

**Influence of Post Edits on Watermark Detection.**    While our primary goal is to design combinatorial patterns that enable more effective edit localization, it is also important to ensure that the underlying watermark remains reliably detectable. We have demonstrated the watermark detectability in Figure 6 using fully watermarked texts. In Figure 8 below, we present some empirical evidence on the influence of post-generation edits on the magnitude of watermark detection statistics, under varying watermarking strengths.

As expected, the watermark detection statistics decrease after post-generation edits, with larger decreases observed for longer edit lengths and more complex patterns. Moreover, for the simplest combinatorial pattern $AB$, the degradation in detection statistics is comparable to—or slightly smaller than—that of the two baseline watermarking approaches. This indicates that the combinatorial pattern-based watermarking maintains at least the same level of robustness to post-generation edits, and may even offer improved resilience in certain scenarios. A more comprehensive analysis of robustness, including theoretical aspects such as detection threshold and sensitivity to different edit types, is left for future work. In the following subsection B.3, we present preliminary theoretical insights to illustrate general trends and motivate further study.

**Mixed Edit Types.**    To evaluate the robustness of our watermark under diverse editing scenarios, we experimented with randomized non-contiguous edits. We sampled 1,200 prompts from Wiki-Text2 and generated 1,200 watermarked texts, each 175 tokens long, using LLaMA-2-13B embedded with the AB combinatorial pattern watermark with $\delta = 5.8$, following the setup described in Section 4. Each text was subjected to edits at 4, 6, or 8 random positions, where each edit modified 3 consecutive tokens through insertion, deletion, or replacement operations–resulting in up to 24

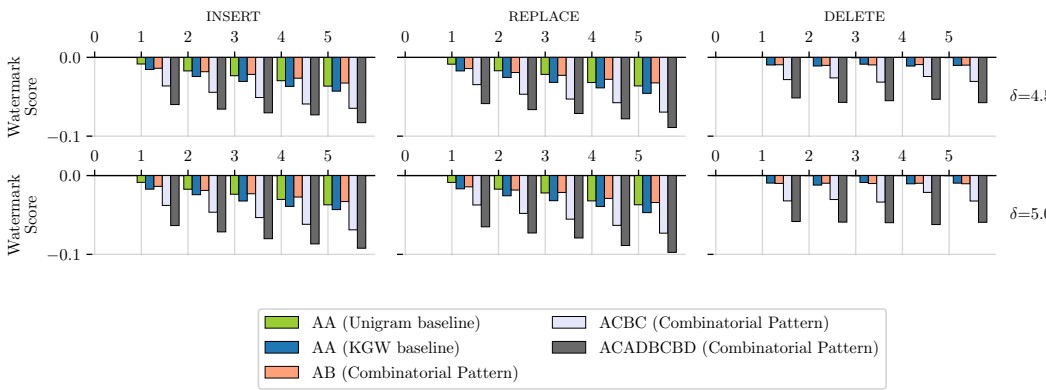

Figure 8: Watermark score impact vs edit type and length under different watermarking strengths $\delta$. The y-axis shows the watermark score difference $|\tilde{s}|_D$ - $|s|_D$, where $s$ and $\tilde{s}$ denote the original watermarked text and the edited text, respectively. The negative values on the y-axis indicate the watermark detection statistics decrease after edits

edited tokens per sample. These experiments introduce mixed edit types to better reflect realistic adversarial scenarios. Fixing the Type-I error below 3%, we summarize the key results in Table 4.

Table 4: Detection accuracy under randomized mixed-type local edits.

| Edit Count | 12 tokens | 18 tokens | 24 tokens |
|---|---|---|---|
| Detection Accuracy | 0.9650 | 0.9675 | 0.9617 |

**Meaningful Edit Detection / Misinformation Spoofing Attack.** To test the robustness of our watermark against realistic threats, we simulated a **targeted misinformation attack**. We began by generating 139 texts, each 100 tokens long, using OPT-2.7b, embedded with the AB combinatorial pattern watermark with $\delta = 4.5$. We then tasked the Gemini API to introduce small but harmful edits to the texts one by one using the following prompt:

> **Gemini-2.5-Flash Prompt**
>
> You are an expert on history, facts, and journalism.
> I will give you a text, and your task is to modify part of it so it gives clear misinformation.
> - Your change MUST significantly alter the meaning of the text.
> - Only change 6 words or less, and leave the rest of the text intact.
> - DO NOT ADD extra formatting, emphasis, punctuation, bolds, or italics.
> - Only respond with the modified text. Nothing else.

A human evaluation of a sample of 50 modified texts found that 90% indeed did contain clear misinformation. Finally, we applied our edit detection algorithm to these adversarially modified texts, and evaluated the edit detection accuracy. Table 5 shows the detection accuracy results, grouped by the number of tokens edited.

Table 5: AB (Combinatorial Pattern): Edit detection accuracy under a misinformation attack. The table buckets texts by the number of "Tokens Edited", reports how many "Texts" fall under each bucket, and the corresponding "Edit Detection Accuracy".

| Tokens Edited | Edit Detection Accuracy (%) | Texts |
|---|---|---|
| 1 | 81.4% | 59 |
| 2 | 90.9% | 44 |
| 3-9 | 91.7% | 36 |

This trend is consistent with our random edit simulations: our algorithm is able to detect and localize edits, and its accuracy increases when more consecutive tokens are modified.

**Comparing with Previous Methods.** While no prior work directly addresses our specific post-watermark edit detection task, several recent studies on watermarked segments localization are related. For example, Zhao et al. (2025) focuses on detecting watermarked segments, and its detection algorithm is compatible with multiple watermarking schemes. We therefore adapt the Adaptive Online Locator algorithm from (Zhao et al., 2025) to our edit-detection task by treating the complement of the detected watermarked tokens as the set of detected edited tokens. We then compare the edit detection accuracy of our proposed method with that of the Adaptive Online Locator in Table 6. Specifically, we provide text watermarked with the AB pattern ($\delta = 2.5$) and assign each token a score $s_t$ equal to $I_w(t)$ from equation 4. We then input this $s_t$ score to the Aligator algorithm from Zhao et al. (2025), which would localize the watermark. To localize the edits instead, we reverse the detection condition from "greater than threshold" to "less than threshold". We evaluate the detected edit positions using our tolerance-based token-level accuracy. It can be seen from the results in Table 6 that our proposed method achieves a substantially higher edit detection accuracy under the same FPR under various edit counts ranging from 1 to 6.

Moreover, WaterSeeker (Pan et al., 2025) is another recent method on watermarked-segment detection, and we comment on its applicability to our setting here. For WaterSeeker, the base algorithm for watermark detection fails to identify any watermarked segments shorter than 20 tokens when using KGW with $\delta = 2.5$. Modifying the algorithm to detect edited positions within a watermarked text performs even more poorly. Therefore, we do not include WaterSeeker in our comparison table below.

Table 6: Comparison of the edit detection accuracy of our detection algorithm and the Adaptive Online Locator algorithm in Zhao et al. (2025), for detecting edits made to watermarked text via AB Pattern on the model facebook/OPT-1.3b. Here $\tau$ denotes the detection threshold.

| Edited Tokens | | AB ($\delta$=2.5) | | | Adaptive Online Locator ($\delta$=2.5) | | |
|---|---|---|---|---|---|---|---|
| Type | Count | $\tau$ | FPR (%) | Accuracy (%) | $\tau$ | FPR (%) | Accuracy (%) |
| INSERT | 1 | 0.4375 | 3.5278 | 39.0625 | 0.77 | 3.5217 | 17.9688 |
| INSERT | 2 | 0.4375 | 3.5400 | 55.4688 | 0.77 | 3.8452 | 23.0469 |
| INSERT | 3 | 0.4375 | 3.5461 | 64.3229 | 0.77 | 4.0100 | 29.4271 |
| INSERT | 4 | 0.4375 | 3.5522 | 74.4141 | 0.77 | 4.4678 | 47.6562 |
| INSERT | 5 | 0.4375 | 3.5156 | 69.5312 | 0.77 | 4.0955 | 46.4063 |
| INSERT | 6 | 0.4375 | 3.5645 | 70.3125 | 0.77 | 4.5959 | 54.9479 |
| REPLACE | 1 | 0.4375 | 3.4510 | 55.4688 | 0.77 | 3.8509 | 19.5312 |
| REPLACE | 2 | 0.4375 | 3.5094 | 56.2500 | 0.77 | 4.1419 | 25.7812 |
| REPLACE | 3 | 0.4375 | 3.5500 | 51.0417 | 0.77 | 4.3938 | 32.2917 |
| REPLACE | 4 | 0.4375 | 3.4841 | 67.5781 | 0.77 | 4.4166 | 46.2891 |
| REPLACE | 5 | 0.4375 | 3.5506 | 63.2812 | 0.77 | 4.3890 | 44.0625 |
| REPLACE | 6 | 0.4375 | 3.4196 | 79.6875 | 0.77 | 4.8796 | 54.1667 |
| DELETE | 1 | 0.4375 | 3.5280 | 14.8438 | 0.77 | 4.0799 | 14.8438 |
| DELETE | 2 | 0.4375 | 3.4375 | 56.2500 | 0.77 | 3.9437 | 14.8438 |
| DELETE | 3 | 0.4375 | 3.5408 | 17.1875 | 0.77 | 4.0449 | 16.4062 |
| DELETE | 4 | 0.4375 | 3.5633 | 50.7812 | 0.77 | 4.1540 | 19.5312 |
| DELETE | 5 | 0.4375 | 3.4900 | 21.8750 | 0.77 | 3.9959 | 18.7500 |
| DELETE | 6 | 0.4375 | 3.5640 | 50.0000 | 0.77 | 4.2485 | 14.0625 |

### B.3 A SENSITIVITY ANALYSIS ON THE WATERMARK DETECTABILITY AND ROBUSTNESS

We provide some analysis on the impact of post-generation edits on the watermark detection statistic. In general, the added edits will decrease the watermark detection statistics, as demonstrated in Figure 8, thus decreasing the watermark detectability.

Recall the watermark detector

$$|\mathbf{s}|_D \;=\; \frac{1}{N} \sum_{t=1}^{N} I_w(t), \quad N = T - w + 1,$$

where $w$ is the sliding-window length, $I_w(t) \in \{0,1\}$ indicates whether the window $s^{(t:t+w-1)}$ matches the watermark pattern, $T$ is the text length, and let $M = \sum_t I_w(t) = N \cdot |\mathbf{s}|_D$ be the total number of windows that match the pattern.

Note that a token at absolute position $u \in \{1, \ldots, T\}$ belongs to the $w$ windows whose *starting* indices lie in

$$\big\{\, u - w + 1,\, u - w + 2,\, \ldots,\, u \,\big\} \cap [N].$$

Hence *any single-token perturbation can flip at most $w$ indicators $I_w(t)$.*

In the following, we use $S_{\mathrm{ins}}$, $S_{\mathrm{del}}$, $S_{\mathrm{rep}}$ to denote the token numbers in insertions, deletions, replacements. And let $\tilde{I}_w(t)$ denote the indicator after editing. Meanwhile, we use $\Delta_\bullet$ to denote the worst-case loss of the number of matched windows attributable to the corresponding edit type $\bullet$. We first consider three cases separately.

- **Replacements**. Replacements alter content but keep length fixed, so $N$ remains unchanged. Moreover, each new token can break at most $w$ windows. Therefore for $S_{\mathrm{rep}}$ replacements, the worst-case loss of matched windows is $\Delta_{\mathrm{rep}} = \min\{wS_{\mathrm{rep}}, M\}$. For the resulted edited text $\tilde{\mathbf{s}}$, the watermark detection statistics after edits thus become

$$|\tilde{\mathbf{s}}|_D \geq \frac{M - \Delta_{\mathrm{rep}}}{N}.$$

- **Insertions**. Adding $S_{\mathrm{ins}}$ tokens grows length to $T + S_{\mathrm{ins}}$, so the window count increases from $N$ to $N + S_{\mathrm{ins}}$. For $S_{\mathrm{ins}}$ insertions, we have the worst-case loss of matched windows is $\Delta_{\mathrm{ins}} = \min\{wS_{\mathrm{ins}}, M\}$. For the resulted edited text $\tilde{\mathbf{s}}$, this yields

$$|\tilde{\mathbf{s}}|_D \;\geq\; \frac{M - \Delta_{\mathrm{ins}}}{N + S_{\mathrm{ins}}}.$$

Since $S_{\mathrm{ins}}$ appears in the denominator, the worst-case statistics after insertions degrade *faster* than replacements (which leave $N$ unchanged).

- **Deletions**. Removing $S_{\mathrm{del}}$ tokens shortens the text, so the window count decreases from $N$ to $N - S_{\mathrm{del}}$. Again at most $w$ of the indicators $I_w(t)$ can flip, giving $\Delta_{\mathrm{del}} = \min\{wS_{\mathrm{del}}, M\}$. The watermark detection statistics after edits thus satisfy

$$|\tilde{\mathbf{s}}|_D \;\geq\; \frac{M - \Delta_{\mathrm{del}}}{N - S_{\mathrm{del}}}.$$

To summarize, each single-token edit can disrupt at most $w$ windows as shown in the following table.

| Edit type | Lost matches | Window count change |
|---|---|---|
| Insertion | $\leq w$ | $+1$ |
| Deletion | $\leq w$ | $-1$ |
| Replacement | $\leq w$ | $0$ |

Collecting the individual effects and clipping at zero yields the deterministic worst-case bound

$$|\mathbf{s}_{\mathrm{edited}}|_D \;\geq\; \frac{M - w\big(S_{\mathrm{ins}} + S_{\mathrm{del}} + S_{\mathrm{rep}}\big)}{N + S_{\mathrm{ins}} - S_{\mathrm{del}}}, \tag{6}$$

here $\mathbf{s}_{\mathrm{edited}}$ denotes the resulting text after all edits. The numerator loses up to $w$ matches per corrupted token; the denominator is stretched by insertions and contracted by deletions, remaining unchanged for replacements.

**Interpreting the bound**. From the worst-case lower bound in equation 6, it can be seen that if one wishes to tolerate at most $\big(S_{\mathrm{ins}}, S_{\mathrm{del}}, S_{\mathrm{rep}}\big)$ benign edits, we can plug those values into equation 6

and set the decision threshold $\tau_d$ no greater than the resulting lower bound. This guarantees that if the watermark is detectable before the edit, then it can also be detected after $\left(S_{\text{ins}}, S_{\text{del}}, S_{\text{rep}}\right)$ edits. Moreover, since a single insertion, deletion, or replacement can disrupt at most $w$ matching windows, the worst-case degradation ($|\mathbf{s}|_D - |\mathbf{s}_{\text{edited}}|_D$) grows linearly with $w$, and thus a smaller window size yields smaller worst-case degradation. However, a smaller window size might be less effective in detecting edits, so there exists some tradeoff in window size selection, and we generally use a larger window for longer patterns in this work.

## C  COMPLEXITY ANALYSIS

Let $T$ be the length of the text in tokens and $w$ be the length of the sliding window for detection. The complexity of our detection metrics, defined in equation 4, is as follows. First, we analyze the complexity of the naive implementation.

- **Single Window Score Complexity** $f(I_w(t))$: To calculate the score for a single window at position $t$, we compare the token window of length $w$ against all $w$ possible cyclic shifts of the watermark pattern. Each comparison involves $w$ token-wise operations, taking $O(w)$ time. Therefore, the total time complexity for one window is $O(w^2)$.
- **Detection Score Complexity** $f(|\mathcal{S}|_D)$: This score requires computing $I_w(t)$ for every possible window in the text. There are $T - w + 1$ such windows. The total complexity is thus $O(T) \cdot O(w^2) = O(Tw^2)$.
- **Edit Score Complexity** $f(|\mathcal{S}|_E(t))$: The edit score at a position $t$ requires $w$ evaluations of the $I_w$ function. The complexity is therefore $w \cdot O(w^2) = O(w^3)$. To compute this for all $T$ positions in the text, the total complexity becomes $O(Tw^3)$.

The naive approach can be optimized using techniques such as rolling hashes. A rolling hash allows the hash of a new window (e.g., from token $t$ to token $t + 1$) to be calculated in $O(1)$ time from the previous window's hash. This would reduce the *amortized* complexity of computing a window's hash to $O(1)$. Then using a hash-table, we can look up if there are any matching shifts in $O(1)$. Consequently, the complexities would become:

$$f(I_w(t)) = O(1), \quad f(|\mathcal{S}|_D) = O(T), \quad \text{and} \quad f(\{|\mathcal{S}|_E(t)\}_{t=1}^T) = O(Tw).$$

In this work, we use the naive implementation, as the window size $w$ is small and fixed in all our experiments.

## D  EXAMPLES OF GENERATED TEXT

To complement the quantitative results, we also provide qualitative examples of texts generated under KGW watermarking and our proposed AB pattern at several watermark strengths $\delta$, under two prompts in Table 7 and 8. These examples illustrate that surface-level fluency and semantic content are largely preserved even as $\delta$ increases, indicating that increasing $\delta$ does not meaningfully degrade text quality.

## LLM USAGE

We utilize open-source LLMs to generate watermarked and edited texts, on which we evaluate watermark detection and edit detection tasks. LLMs are also used for polishing the writing slightly. No ideas, analyses, or discoveries are contributed by LLMs.

Table 7: Comparison of generated examples under KGW watermarking and our proposed AB pattern across different watermark strengths $\delta$.

| | KGW | AB |
|---|---|---|
| Prompt: In 1950 Alan Turing published a landmark paper in... | | |
| $\delta$=0.0 | ...which he introduced the concept of a Universal Turing Machine (UTM). The UTM is a hypothetical machine that can be programmed to simulate the logic of any other computer. It is often referred to as a universal computer or a Turing machine. The UTM is a theoretical | ...which he introduced the concept of a Universal Turing Machine (UTM). The UTM is a hypothetical machine that can be programmed to simulate the logic of any other computer. It is often referred to as a universal computer or a Turing machine. The UTM is a theoretical |
| $\delta$=2.5 | ...which he introduced the idea of a machine that could perform any computable function. This machine, known as the universal Turing machine, has become one of the most important abstractions in all of mathematics. In this book, S. Barry Cooper tells the story of the universal Turing machine and its role in the | ...which he proposed a model of machine intelligence. He argued that machines could be programmed to perform tasks that were previously considered the domain of humans. The idea that machines could think like humans has been a central theme in artificial intelligence since then. Turings proposal was based on the idea that machines could be program |
| $\delta$=3.5 | ...the field of artificial intelligence. The paper was called Computer Machinery and Intelligence, and it introduced the idea of a test for artificial intelligence. The test, known as the Turing Test, is now considered the gold standard for evaluating artificial intelligence. In this blog post, we | ...which he proposed a model of machine intelligence. He argued that machines could be programmed to perform human-like intellectual tasks if their memory were unlimited, if they could examine pieces of evidence, if they could use general rules for combining pieces of evidence, if they could use general rules for combining pieces of evidence, if |
| $\delta$=4.5 | ...the field of artificial intelligence. The paper was called Computer Machinery and Intelligence, and it introduced the idea of a test for artificial intelligence. The test, known as the Turing Test, is now considered the gold standard for evaluating artificial intelligence. In this blog post, we | ...which he proposed a model of machine intelligence. He argued that machines could be programmed to perform human-like tasks, and that they could be tested by giving them a set of simple rules and then seeing how well they fared at solving the kinds of problems that humans find challenging. Turings ideas |
| $\delta$=5.5 | ...the field of artificial intelligence. The paper was called Computer Machines and Intelligence, and it introduced the idea of a test that could be applied to any machine, whether it was a human or a robot, that could tell us whether it was actually intelligent, or not. The test was called | ...which he proposed a model of machine intelligence. He argued that machines could be programmed to perform human-like tasks, and that they could be tested by giving them a set of simple rules and then seeing how well they imitated human performance on an IQ test. Turings idea was that machines could |

Table 8: Comparison of generated examples under KGW watermarking and our proposed AB pattern across different watermark strengths $\delta$.

| | KGW | AB |
|---|---|---|
| Prompt: Several philosophers had strong objections to the claims being made by AI... | | |
| $\delta$=0.0 | ...researchers. For example, Hubert L. Dreyfus, a professor of philosophy at the University of California, Berkeley, wrote in his book What Computers Still Can't Do: A Critique of Artificial Reason (1992) that "artificial intelligence is a | ...researchers. For example, Hubert L. Dreyfus, a professor of philosophy at the University of California, Berkeley, wrote in his book What Computers Still Can't Do: A Critique of Artificial Reason (1992) that "artificial intelligence is a |
| $\delta$=2.5 | ...proponents, such as John Searle, who famously argued in his Chinese Room Experiment that no computer could understand Chinese, even though it may be able to mimic the behavior of someone who understands the language. In the 1980s, the field of cognitive science was | ...systems and their proponents. One of the most prominent was John Searle. In his 1980 paper Minds, brains and programs, he challenged the claim that thinking is a computational process. He argued that a computer could never be said to be thinking in the same sense that a person |
| $\delta$=3.5 | ...scientists. In 1984, John Searle, professor of philosophy and cognitive science at the University of California, Berkeley argued in his Chinese Room Experiment that no computer could understand a human language. The argument goes as follows. A computer program is set up in a room | ......systems that they could think, feel, and even have consciousness. One of the loudest voices in this debate was John Searle. In his 1980 paper Minds, brains, and programs Searle wrote: [T]he fact that a computer can do a certain |
| $\delta$=4.5 | ...scientists. In 1843, Charles Babbage, the inventor of the first computer, said: "I am not the first of the inventors but I am the first of the practitioners, inasmuch as the application of my system has been begun and it will be | ...systems that they could think, feel, and even have consciousness. One of the loudest voices in this debate was the late Daniel Dennett, a professor of philosphy at Tufts. In his 1987 book The Intentional Stance, he coined the term the intentional |
| $\delta$=5.5 | ...scientists. In 1843, Charles Babbage, the inventor of the first computer, said: "I am not the first of the inventors but I am the first of the practitioners, inasmuch as my machine exists and works." (Babbage, | ...systems that they could think, feel, and even have consciousness. One of the loudest voices in this debate was the late John Searle. In his 1980 paper titled, Minds brains and program, Searle makes the argument that no matter how complex or sophistic |

