# OpenReview forum: "Detecting Post-generation Edits to Watermarked LLM Outputs via Combinatorial Watermarking"
_ICLR.cc/2026/Conference — Submitted to ICLR 2026_

### Official Review · Reviewer_Jmfd · 2025-10-23

**Soundness:** 2
**Presentation:** 2
**Contribution:** 2
**Rating:** 2
**Confidence:** 4

**Summary:**

This paper introduces a novel hard-watermarking scheme designed to detect and localize edits in text generated by large language models. The core idea is to embed a predefined pattern (e.g., A-B-A-B) into the text during generation and subsequently detect edits by identifying disruptions to this pattern using a sliding window approach. While the problem of edit localization is highly relevant and important, I have several major concerns regarding the experimental methodology, the practical viability of the proposed method, and the lack of comparison to relevant baselines. These concerns prevent me from recommending acceptance in its current form. The paper would require a major revision to address these issues.

**Strengths:**

The primary strength of this work lies in its focus on a critical and practical problem. As watermarking becomes a standard for identifying machine-generated text, the ability to not only detect the presence of a watermark but also to precisely localize subsequent human edits is of great importance for content attribution, tracking modifications, and ensuring information integrity. The authors are commended for tackling this challenging task of edit localization.

**Weaknesses:**

1. **Lack of Clarity in Experimental Results**: The main experimental results for edit detection accuracy are presented solely in Figure 5. However, the visualization makes it difficult to assess the performance precisely. The y-axis ticks are too sparse, preventing a clear and direct comparison of the exact performance values between different methods and conditions. For a rigorous evaluation, graphical representations should be supplemented with precise data. I strongly recommend that the authors provide a table that reports the exact numerical values for edit detection accuracy for all methods and conditions shown in Figure 5.

2. **Severe Impact on Text Quality**: The proposed method is a "hard" watermark, which restricts the sampling vocabulary at each position to a specific pre-determined list of tokens. This approach is known to have a significant negative impact on the quality (fluency, coherence, and factual correctness) of the generated text. This concern is amplified by the choice of delta=5.8 for the main experiments in Figure 5. This is an extraordinarily high value for the logit bias parameter. Standard practice in the field often uses a delta around 2.0, as higher values can severely degrade text quality to the point of being nearly unreadable. The paper currently lacks any evaluation of the text quality under such a strong watermark, which is a critical omission. Without this, the reported detection accuracy is meaningless, as it may come at the cost of rendering the text unusable.

3. **Missing Comparison to Relevant Baselines**: The task of edit localization is conceptually similar to the task of identifying watermarked segments within a mixed-source document (i.e., a document containing both watermarked and non-watermarked text). The paper fails to discuss or compare against recent, relevant work in this area, which is a significant oversight. A thorough discussion and empirical comparison with existing segment detection methods are necessary to properly contextualize the contributions of this paper and to demonstrate its advantages over prior art. The authors should consider comparing their method against baselines such as:

[1] Efficiently Identifying Watermarked Segments in Mixed-Source Texts
[2] WaterSeeker: Pioneering Efficient Detection of Watermarked Segments in Large Documents

**Questions:**

1. Could the authors provide a clear justification for choosing delta=5.8 for the experiments in Figure 5? Why was such a strong bias necessary? Furthermore, it is crucial to demonstrate the quality of the text generated with this setting. I would expect to see standard automatic metrics like Perplexity (PPL) and, if possible, performance on downstream NLP tasks to quantify the trade-off between detection accuracy and text utility. A qualitative analysis with generated examples would also be highly beneficial.

2. The evaluation in Figure 5 assesses the detection of "pure" edit types (i.e., only inserts, only deletes, or only replacements) in isolation. However, real-world edits are often complex and involve a combination of these operations (e.g., rephrasing a sentence involves both deletions and insertions). Could the authors provide an evaluation of the method's robustness in more realistic, mixed-edit scenarios? How does the detection accuracy hold up when multiple types of edits are present within the same text segment?

---

> ### Author Response · Authors · 2025-11-22
>
> We thank the reviewer for their thoughtful and constructive feedback. We are especially grateful for the recognition of the strengths of our work, including “the problem of edit localization is highly relevant and important”, “a critical and practical problem”, “The authors are commended for tackling this challenging task of edit localization”.
>
> Below we address the specific concerns raised.
>
> **Q:** *"Provide a table that reports the exact numerical values for edit detection accuracy for all methods and conditions shown in Figure 5."*
>
> **A:** Thank you very much for the helpful suggestion. We have now added a table (Table 2 in Appendix B.2) that reports the exact numerical values of edit detection accuracy for all methods and conditions shown in Figure 5. This table clearly shows the quantitative comparison and thus the advantage of our proposed method in detection accuracy. Moreover, as inspired by your other suggestions, we also include a comprehensive Table 3 in Appendix B.2 that reports the edit detection accuracy under other smaller delta values.
>
> **Q:** *"Justifications of delta=5.8, and evaluation of the text quality under the proposed watermarking.*"
>
> **A:** Thank you for the thoughtful suggestions. First, we chose delta=5.8 in Figure 5 based on empirical observations that they offer a favorable trade-off between output quality and edit detection accuracy. We would like to emphasize that our method is compatible with smaller delta values as well. We have now included a comprehensive Table (Table 3 in Appendix B.2) that reports the edit detection accuracy under other smaller delta values. Generally speaking, smaller $\delta$ values yield better output quality and still maintain good watermarking detectability, but may weaken edit detection performance.
>
> Second, we acknowledge that the combinatorial pattern-based watermarking may impact text quality, but we would like to kindly note that the impact is (i) a fundamental *trade-off between edit detectability and output quality* in our design—longer patterns improve edit detection but may reduce output quality. This trade-off is *tunable* via pattern design and watermarking strength, allowing users to balance robustness and quality based on real needs; (ii) and the impact is not severe to the best of our observations. To give a qualitative analysis of the generated examples, we have now added examples of the generated texts, under 2 different prompts and different delta values, in Appendix D (Tables 6 and 7).
>
> **Q:** *"A thorough discussion and empirical comparison with existing segment detection methods.*"
>
> **A:** Thank you for the suggestions. We have added discussions in the revised Related Work section on these references on watermarked segment detection. In summary, these approaches are designed to detect long, contiguous watermarked spans in mixed-source documents and assume that the underlying watermarked text remains largely unmodified. In contrast, our setting focuses on post-generation local edits—small insertions, deletions, and substitutions that corrupt an originally watermarked sequence—where segment-detection frameworks are not naturally applicable.
> Nonetheless, we agree that an empirical comparison is valuable. We are currently adapting these detectors to our simulated post-generation edit setting and running additional experiments. We will update the results once the evaluations are completed.

---

> > ### Author Response · Authors · 2025-11-22
> >
> > **Q:** *"Evaluation of the method's robustness in more realistic, mixed-edit scenarios.*"
> >
> > **A:** Thank you for raising this point. We refer the reviewer to the Table 5 and the newly added Table 4 in Appendix B.2. Due to space limit, we only presented the results under canonical edits in the main paper, but we did conduct additional experiments under spoofing-attack-type edits generated by Gemini, in which we didn’t limit the edit type.
> >
> > Specifically, to test the robustness of our watermark against realistic threats, we simulated a targeted misinformation attack. We began by generating 139 texts, each 100 tokens long, using OPT-2.7b. We then tasked the Gemini API to introduce small but harmful edits by giving it the prompt “...your task is to modify part of it so it gives clear misinformation…”. Finally, we applied our edit detection algorithm to these adversarially modified texts, and observed the edit detection accuracy to be at least 81.4% and more than 90% for most edits. The results can be found in Table 5.
> >
> > Furthermore, we also conducted additional experiments using a larger model (LLaMA-2-13B) with controllable randomized extensive edits. We use 1200 prompts from WikiText2 and generated 1,200 watermarked texts (*175 tokens each*) using LLaMA-2-13B with the AB pattern, following the setup in Section 4. Each text was edited at *4, 6, or 8* random positions, with each edit modifying *3 consecutive tokens* via insertion, deletion, or replacement—totaling *up to 24 edited tokens* per sample. Therefore, these experiments introduce mixed edit types to better reflect realistic scenarios. Fixing the Type-I error below 3%, we summarize the key results in the newly added Table 4 in the Appendix. The detection accuracy are consistently above 95%.
> >
> > We hope that these results strengthen the empirical support for our method under more challenging and realistic edit conditions.
> >
> > Finally, we thank the reviewer again for the thoughtful feedback and hope our responses and revions have clarified the key points. Please let us know if any further questions arise, and we sincerely appreciate the opportunity to clarify and strengthen our submission.

---

> > > ### Comment · Reviewer_Jmfd · 2025-11-28
> > > **Thanks for your clarifications. I have one more question.**
> > >
> > > Thanks for your clarifications. I have one more question. Regarding the text quality evaluation, there are only some examples provided. Could you provide some statistical results, such as PPL and performance on some downstream tasks (e.g. machine translation, code generation, summarization)?

---

> > > ### Author Response · Authors · 2025-11-28
> > >
> > > Thank you again for your suggestion to **compare with existing watermark–segment detection algorithms**. As noted in our initial response, we have conducted these experiments and incorporated the results and discussion into the revision. We added a new paragraph in Appendix B.2 (highlighted in blue in the revision) and a new Table 6 that evaluate/discuss two relevant baselines: [1] Efficiently Identifying Watermarked Segments in Mixed-Source Texts and [2] WaterSeeker: Pioneering Efficient Detection of Watermarked Segments in Large Documents.
> > >
> > > For [1], we adapted their Adaptive Online Locator algorithm to our edit-detection setting by treating the complement of detected watermarked tokens as detected edited tokens. The full numerical comparison is presented in Table 6, where our method consistently achieves substantially higher edit detection accuracy at comparable (or slightly lower) False Positive Rate (FPR) levels across a range of edit counts. These results confirm that our proposed algorithm yields substantially stronger abilities of post-generation edit detection and localization. For [2], as detailed in the added paragraph, the base WaterSeeker algorithm is unable to reliably detect watermarked segments shorter than 20 tokens under KGW under our experienmetns at  $\delta=2.5$, and we observe that its performance further deteriorates when adapted to edit detection. For this reason, and as explained in the manuscript, we do not include WaterSeeker in the numerical comparison table.
> > >
> > > Importantly, we believe the performance observed in these baselines stems from a difference in problem formulation: watermark-segment detection methods are designed to detect contiguous spans of watermarked text, and thus typically rely on sliding-window procedures optimized for that objective. In contrast, our task is edit detection, where edits may occur sparsely throughout a watermarked sequence. Consequently, our pattern-based detector, which is designed specifically for such edit setting, is inherently more suitable for this task and tends to achieve stronger edit detection performance.
> > >
> > > We hope that these additional results and the expanded discussion in Appendix B.2 address the reviewer’s request and clarify how our method compares to existing watermark-segment detection approaches.

---

> > > > ### Comment · Reviewer_Jmfd · 2025-11-28
> > > > **Thanks for your explanation, but my following-up question is about text quality.**
> > > >
> > > > Thanks for your explanation, but my following-up question is about text quality. Please refer to this:
> > > >
> > > > > Thanks for your clarifications. I have one more question. Regarding the text quality evaluation, there are only some examples provided. Could you provide some statistical results, such as PPL and performance on some downstream tasks (e.g. machine translation, code generation, summarization)?

---

> > > > > ### Author Response · Authors · 2025-11-29
> > > > >
> > > > > Thank you again for the helpful follow-up question regarding text quality. Since receiving your follow-up comment, we have been working on adding **the perplexity (PPL) results to Table 3** in the revised paper, to make the evaluation more comprehensive and to jointly reflect text quality alongside edit-detection performance.
> > > > >
> > > > > As shown in the newly revised Table 3 and the newly added text nearby (highlighted in blue), we now report PPL values for our watermarking method under several pattern designs and watermarking strength ($\delta$). For the reviewer’s convenience, we provide below a compact excerpt of the PPL portion of Table 3 (full table is available in Table 3 in the revision). The results indicate that under the simplest AB pattern, our method achieves perplexity comparable to the Unigram and KGW baselines across a range of watermarking strengths. Together with the qualitative examples included in Tables 7 and 8, these results suggest that the impact on text quality is minimal.
> > > > >
> > > > > | Pattern     | δ=2.5 | δ=3.5 | δ=4.5 | δ=5.5 |
> > > > > |-------------|-------|-------|-------|-------|
> > > > > | Unigram     | 6.97  | 9.36  | 13.31 | 19.78 |
> > > > > | KGW         | 7.00  | 9.38  | 13.29 | 19.61 |
> > > > > | AB (ours)          | 6.96  | 9.25  | 12.97 | 18.89 |
> > > > > | AB (rand) (ours)  | 6.96  | 9.27  | 13.00 | 18.96 |
> > > > >
> > > > > More complex patterns can slightly increase perplexity, which reflects the fundamental trade-off between edit detectability and output quality. This trade-off is tunable: by adjusting the pattern structure and watermarking strength, practitioners can balance text quality and detection performance according to the needs of the application. Generally speaking, smaller watermarking strength $\delta$ and shorter patterns are sufficient to achieve a good balance between text quality and edit-detection performance; higher watermarking strength and longer patterns can be used when near-perfect edit detection is required, but may come with a slight reduction in text quality.
> > > > >
> > > > > We hope that these newly added results and examples address the reviewer’s concerns about text quality.

---

### Official Review · Reviewer_gEsA · 2025-11-01

**Soundness:** 3
**Presentation:** 3
**Contribution:** 3
**Rating:** 6
**Confidence:** 4

**Summary:**

This paper introduces a new task, post-generation edit detection, aimed at identifying and localizing edits made to watermarked LLM outputs. To address this, the authors propose a combinatorial pattern-based watermarking framework, where the vocabulary is partitioned into multiple subsets and a deterministic cyclic pattern governs token selection during generation. The resulting watermark allows both global detection (via pattern alignment) and local edit localization (via violations of the expected token pattern). Experiments using LLaMA-2-7B and OPT-1.3B demonstrate effective localization of token-level edits (insertions, deletions, and replacements) while maintaining detectability comparable to prior methods such as KGW and Unigram watermarking.

**Strengths:**

* The introduced task is novel and interesting, and the paper is among the first to explicitly address detecting and localizing post-generation edits in watermarked text, an important problem for attribution and content integrity.
* The combinatorial watermarking mechanism is intuitive and easy to integrate into existing watermarking pipelines, offering deterministic behavior for analysis.
* Results across different edit types and lengths convincingly demonstrate that combinatorial patterns, particularly longer ones, yield better localization than KGW and Unigram baselines.
* The formalization of edit detection accuracy and false alarm rate provides a useful evaluation framework that could benefit future work.

**Weaknesses:**

* The proposed combinatorial watermark shares conceptual similarity with earlier multi-bit or multi-channel watermarking schemes that encode structured token groupings across vocabulary subsets (e.g. Yoo et al., NAACL 2024). However, such works are not cited or compared. Acknowledging and differentiating from these prior frameworks would strengthen the novelty claim.
* While detecting post-generation edits is novel, the core watermarking method (deterministic pattern over token partitions) feels like a direct extension of prior list-based or multi-bit watermarking. The incremental innovation lies primarily in the application to edit localization rather than in the watermarking technique itself. An elaboration of any specific benefits conferred by this cyclical pattern-based scheme versus partitioning more arbitrarily into multiple vocabulary lists would help clarify the contribution.
* All experiments use short, synthetic WikiText prompts. A small test on more diverse corpora or realistic post-edit scenarios (e.g., paraphrasing, stylistic revision) would better establish generality.

**Questions:**

* Could the authors clarify whether their combinatorial watermark is equivalent or similar in practice to a multi-bit encoding over token indices, and if so, how it differs from earlier multi-bit watermark designs?
* Section 4 states that watermark generation “takes approximately seven seconds per batch” but does not compare this to the unwatermarked generation time. An addition of the wall-time or throughput (tokens/s) for that case would make it more clear whether the watermarking method introduces notable latency or computational overhead.
* Figure 5 is fairly difficult to interpret (a lot of the values basically look like 1.0, and the extent of finer improvements is unclear). A tabular presentation of the results would greatly improve readability and allow easier cross-pattern comparison. Similar for Figure 8 — since that one is in the appendix, even multiple tables, one for each delta value, would be reasonable.
* Does the edit localization performance degrade under semantic-preserving paraphrasing rather than token-level edits?

---

> ### Author Response · Authors · 2025-11-22
>
> We thank the reviewer for their thoughtful and constructive feedback. We are especially grateful for the recognition of the strengths of our work, especially the novelty and importance of the post-generation edit detectioin task.
>
> Below we address the specific concerns raised.
>
> **Q:** *"Comparison with multi-bit encoding over token indices."*
>
> **A:** Thank you for the insightful question and for bringing out this important line of work. While our pattern shares certain similarity with the multi-bit message in Yoo et al. (2024), our combinatorial watermark is not equivalent to the multi-bit encoding over token indices, and they differs fundamentally in terms of the ultimate task and watermarking procedure. We have now added a paragraph in the related-work section clarifying these differences and cited the relevant multi-bit watermarking literature (see the new Multi-bit Watermarking paragraph).
>
> *Different watermarking procedure*: Yoo et al. (2024) propose a multi-bit method in which each generated token is pseudo-randomly assigned to a message position, and the message content at that position is encoded using multi-colored vocabulary partitions. Their goal is user tracing with robustness and low latency. We would like to note that in Yoo et al.’s design, the message itself is arbitrary and externally provided (e.g., user ID, model version), and token positions do not follow a structured or cyclic pattern—each token is allocated independently to random message positions. By contrast, our setting and design are different. We do not embed an arbitrary bitstring. Instead, we construct a combinatorial pattern whose structure is carefully crafted to be sensitive to edits. This pattern is not a message to be decoded, is not associated with model/user identity, and is not arbitrary. And in our watermarking procedure, the pattern is cyclically embedded into the watermarked tokens so that the expected pattern at each token position is predetermined to enable edit detection.
>
> *Different goals*: These multi-bit watermarking methods (e.g., Yoo et al., 2024; Fernandez et al., 2023; Wang et al., 2024) all aim to embed an arbitrary, fixed message—typically a model or user identifier—into the generated text so that the message can later be recovered. Our work aims to detect post-generation token edits, a task that requires internal pattern consistency. Therefore, these multi-bit watermarking methods are evaluated by how accurately this message can be recovered/decoded, while in our case, we need to use edit detection accuracy as our evaluation metric.
>
> **Q:** *"An elaboration of any specific benefits conferred by this cyclical pattern-based scheme versus partitioning more arbitrarily into multiple vocabulary lists would help clarify the contribution."*
>
> **A:** Thank you for the comment. Our cyclical pattern provides concrete benefits that arbitrary multi-list partitioning cannot offer. Taking Yoo et al. (2024) as an example, multi-bit watermarking usually partitions the vocabulary arbitrarily to encode a fixed message, but the resulting token choices are independent across positions, thus they provide no information about where an edit occurred. As a result, those schemes only produce a global message recovery and cannot indicate where an edit has occurred. In contrast, our method introduces a deterministic, cyclical pattern that imposes predictable per-position token-partition expectations. This design ensures that consecutive tokens follow the cyclical pattern and thus insertion, deletion, or replacement immediately violates these local constraints. Therefore, our work is not simply applying multi-bit watermarking to a new task, but introducing a watermarking framework specifically designed for the new demands of post-generation edit detection.

---

> > ### Author Response · Authors · 2025-11-22
> >
> > **Q:** *"Performance under semantic-preserving paraphrasing."*
> >
> > **A:** Thank you for the great question. Semantic-preserving paraphrasing is much harder to detect than token-level edits, and our method, at the current stage, is not designed for such scenario. As a first attempt at post-generation edit localization, we intentionally restrict attention to these token-level modifications, as they are conceptually simple yet still impactful: token-level edits (insertions, deletions, substitutions) can already distort the intended meaning or safety properties of the LLM output. Handling semantic-preserving paraphrasing is outside our current scope but is an interesting direction for future work.
> >
> > We also offer our brief perspective on detecting semantic-preserving paraphrasing. From a spoofing-attack point of view, if a user rewrites the text while fully preserving its meaning, failing to flag such edits is generally acceptable; semantic-level watermarks (e.g., Ren et al., 2024) may still identify the text as LLM-generated even after paraphrasing. From a technical point of view, one could add both semantic-based watermarks and token-level watermarks to the LLM output: if the semantic watermark remains detectable while the token-level pattern is broken, this may indicate heavy semantic-preserving paraphrasing. Exploring such hybrid designs is beyond our current scope but is an interesting direction for future work.
> >
> > Ren, Jie, Han Xu, Yiding Liu, Yingqian Cui, Shuaiqiang Wang, Dawei Yin, and Jiliang Tang. "A robust semantics-based watermark for large language model against paraphrasing." In Findings of the Association for Computational Linguistics: NAACL 2024, pp. 613-625. 2024.
> >
> >
> > **Q:** *"Need more diverse corpora or realistic post-edit scenarios."*
> >
> > **A:** Thank you for raising this point. We refer the reviewer to the Table 5 and the newly added Table 4 in Appendix B.2. Due to space limit, we only presented the results under canonical edits in the main paper, but we did conduct additional experiments under spoofing-attack-type edits generated by Gemini, in which we didn’t limit the edit type.
> >
> > Specifically, to test the robustness of our watermark against realistic threats, we simulated a targeted misinformation attack. We began by generating 139 texts, each 100 tokens long, using OPT-2.7b. We then tasked the Gemini API to introduce small but harmful edits by giving it the prompt “...your task is to modify part of it so it gives clear misinformation…”. Finally, we applied our edit detection algorithm to these adversarially modified texts, and observed the edit detection accuracy to be at least 81.4% and more than 90% for most edits. The results can be found in Table 5.
> >
> > Furthermore, we also conducted additional experiments using a larger model (LLaMA-2-13B) with controllable randomized extensive and mixed-type edits. We use 1200 prompts from WikiText2 and generated 1,200 watermarked texts (*175 tokens each*) using LLaMA-2-13B with the AB pattern, following the setup in Section 4. Each text was edited at *4, 6, or 8* random positions, with each edit modifying *3 consecutive tokens* via insertion, deletion, or replacement—totaling *up to 24 edited tokens* per sample. Therefore, these experiments introduce mixed edit types to better reflect realistic scenarios. Fixing the Type-I error below 3%, we summarize the key results in the newly added Table 4 in the Appendix. The detection accuracy are consistently above 95%.
> >
> > We hope that these results strengthen the empirical support for our method under more challenging and realistic edit conditions.
> >
> > **Q:** *"A tabular presentation of Figure 5."*
> >
> > **A:** Thank you very much for the helpful suggestion. We have now added a table (Table 2 in Appendix B.2) that reports the exact numerical values of edit detection accuracy for all methods and conditions shown in Figure 5. This table clearly shows the quantitative comparison and thus the advantage of our proposed method in detection accuracy. Moreover, we also include a comprehensive Table 3 in Appendix B.2 that reports the edit detection accuracy under other smaller delta values and scenarios.
> >
> > **Q:** *"Generation time comparison between watermarked and unwatermarked."*
> >
> > **A:** Thank you for raising this important point. We have added Table 1 in the revised paper to compare the wall-clock generation time between the unwatermarked model and the watermarked model. As shown in the table, for sequences of 2,048 tokens, the unwatermarked and watermarked versions require 4.22s and 4.33s respectively. The generation times are therefore essentially identical, indicating that our watermarking does not introduce additional computational overhead during text generation.
> >
> > Finally, we thank the reviewer again for the thoughtful feedback and hope our responses and revision have clarified the key points. Please let us know if any further questions arise, and we sincerely appreciate the opportunity to clarify and strengthen our submission.

---

### Official Review · Reviewer_D4Hi · 2025-11-01

**Soundness:** 4
**Presentation:** 4
**Contribution:** 3
**Rating:** 6
**Confidence:** 4

**Summary:**

This paper proposes a combinatorial watermarking method that enables reliable detection and localization of post-generation edits in LLM-generated text through lightweight, non-gradient statistical analysis.

**Strengths:**

1. Addresses an important problem — detecting whether watermarked text has been modified.
2. Easily extendable to multi-label and more complex watermark combinations.
3. Low computational cost and black-box applicability make it highly practical.

**Weaknesses:**

1. Unclear scalability to newer watermarking schemes beyond red–green lists, such as embedding-level watermarks.
2. Experiments should be validated on more advanced, state-of-the-art LLMs.
3. Does not evaluate robustness against stronger post-editing methods like full paraphrasing by other LLMs.

**Questions:**

How is the method scalable towards more watermarking styles and more paraphrasing tools, especially on SOTA LLMs?

---

> ### Author Response · Authors · 2025-11-22
>
> We thank the reviewer for their thoughtful and constructive feedback. We are especially grateful for the recognition of the importance and practical usage of our work.
>
> Below we address the specific concerns/questions raised.
>
> **Q:** *"How is the method scalable towards more watermarking styles?."*
>
> **A:** Thank you for the insightful question. In the current stage, we only consider green-red list type watermarking, because we believe they work naturally for the edit detection task. To clarify, our goal is not to make every watermarking scheme support post-generation edit detection. Instead, we propose a combinatorial, token-level watermark because this structure is inherently well-suited for edit localization: it yields discrete, position-specific pattern states that break immediately under insertions, deletions, or substitutions. Newer embedding-level watermarks operate in hidden space and may not naturally provide such per-token deterministic patterns. In practice, since different watermarks may coexist to serve different purposes, our contribution is focused and is to design a watermarking mechanism that enables edit detection, rather than adapting edit detection to all existing watermark schemes.
>
> **Q:** *"How is the method scalable towards more paraphrasing tools, especially on SOTA LLMs?."*
>
> **A:** Thank you for raising this important point. Although we do not evaluate our method under explicit paraphrasing tools, our approach is designed to remain robust as long as local token-level structure is preserved, a property that holds for many real-world editing behaviors. To give more numerical evidences, we would like to refer the reviewer to Table 5 and the newly added Table 4 in Appendix B.2. Due to space limit, we only presented the results under canonical edits in the main paper, but we did conduct additional experiments under spoofing-attack-type edits generated by Gemini, in which we didn’t limit the edit type.
>
> Specifically, to test the robustness of our watermark against realistic threats, we simulated a targeted misinformation attack. We began by generating 139 texts, each 100 tokens long, using OPT-2.7b. We then tasked the Gemini API to introduce small but harmful edits by giving it the prompt “...your task is to modify part of it so it gives clear misinformation…”. Finally, we applied our edit detection algorithm to these adversarially modified texts, and observed the edit detection accuracy to be at least 81.4% and more than 90% for most edits. The results can be found in Table 5.
>
> Furthermore, we also conducted additional experiments using a larger model (LLaMA-2-13B) with controllable randomized extensive edits. We use 1200 prompts from WikiText2 and generated 1,200 watermarked texts (*175 tokens each*) using LLaMA-2-13B with the AB pattern, following the setup in Section 4. Each text was edited at *4, 6, or 8* random positions, with each edit modifying *3 consecutive tokens* via insertion, deletion, or replacement—totaling *up to 24 edited tokens* per sample. Therefore, these experiments introduce mixed edit types to better reflect realistic scenarios. Fixing the Type-I error below 3%, we summarize the key results in the newly added Table 4 in the Appendix. The detection accuracy are consistently above 95%. We hope that these results strengthen the empirical support for our method under more challenging and realistic edit conditions.
>
> Finally, we also acknowledge that our method is not designed to detect heavy paraphrasing, such as cases where the text has been extensively rewritten while preserving the semantic meaning. Such transformations substantially alter the token-level structure, making them far more challenging to detect for any token-based watermarking scheme. From a spoofing-attack perspective, if a user rewrites the text while fully preserving its meaning, failing to flag such cases is generally acceptable; and semantic-level watermarks may still identify the text as LLM-generated even after paraphrasing. From a technical standpoint, one could combine semantic-based watermarks with token-level patterns: if the semantic watermark remains detectable while the token-level pattern is broken, this may signal heavy paraphrasing. While such hybrid approaches fall outside the scope of this work, we believe they represent a promising direction for future research.
>
> Finally, we thank the reviewer again for the thoughtful feedback and hope our responses and revision have clarified the key points. Please let us know if any further questions arise, and we sincerely appreciate the opportunity to clarify and strengthen our submission.

---

> > ### Comment · Reviewer_D4Hi · 2025-11-27
> >
> > Thanks for addressing my concerns. I think it is a noticeable work and will keep my score.

---

### Official Review · Reviewer_kxy5 · 2025-11-01

**Soundness:** 2
**Presentation:** 3
**Contribution:** 2
**Rating:** 4
**Confidence:** 2

**Summary:**

The paper proposes a novel LLM watermarking scheme that facilitates the detection and localization of post-generation edits to watermarked content. The watermark introduces a repeating cycle of disjoint green lists (tokens whose logits are boosted during generation). By sliding a window over the watermarked text and calculating a local statistic for a token, the scheme can probabilistically check if the token satisfies the cyclic token distributions. Evaluation shows that the scheme can detect edits much more reliably than baselines.

**Strengths:**

- The paper highlights a relevant gap in the watermark literature. The problem of detecting post-generation edits is formulated well.

- The idea of cyclic patterns is effective in detecting non-adversarial edits.The probability analyses (e.g. upper bound of false alarm) add rigor to the method.

- Evaluation results indicate the effectiveness of the edit detection and localization across different LLMs.

**Weaknesses:**

- The method trades off both text quality and the global watermark detection accuracy for the ability to detect edits. The imbalanced scale of this trade will significantly impact the adoption of this technique considering the former qualities’ importance.

- The cyclic and deterministic nature of the watermark seems vulnerable to exploits by a knowledgeable adversary. The baseline watermarking techniques do not have this same problem thanks to their use of hashing.

**Questions:**

- Can an attacker with some knowledge about this watermarking scheme exploit the cyclic nature of the watermark? For example, can they determine the vocab set for each tag? And can they align the edits to the cycle (e.g., only edit tokens at tag 1)?

---

> ### Author Response · Authors · 2025-11-22
>
> We thank the reviewer for their thoughtful and constructive feedback. We are especially grateful for the recognition of the strengths of our work, including “highlights a relevant gap in the watermark literature”, “the problem of detecting post-generation edits is formulated well”, “idea of cyclic patterns is effective”, “probability analyses (e.g. upper bound of false alarm) add rigor”, etc.
>
> Below we address the specific concerns raised.
>
> **Q:** *"Trades off both text quality and the global watermark detection accuracy for the ability to detect edits."*
>
> **A:** First of all, we acknowledge that the combinatorial pattern-based watermarking may impact text quality, but we would like to kindly note that the impact is (i) a fundamental *trade-off between edit detectability and output quality* in our design—longer patterns improve edit detection but may reduce output quality. This trade-off is *tunable* via pattern design and watermarking strength, allowing users to balance robustness and quality based on real needs; (ii) and the impact is not severe to the best of our observations. To give a qualitative analysis of the generated examples, we have now added examples of the generated texts, under 2 different prompts and different delta values, in Appendix D (Tables 6 and 7).
>
> We would also like to emphasize that our method is compatible with smaller delta values as well, thus having less impact on text quality. We have now included a comprehensive Table (Table 3 in Appendix B.2) that reports the edit detection accuracy under other smaller delta values. Generally speaking, smaller $\delta$ values yield better output quality and still maintain good watermarking detectability, but may slightly weaken edit detection performance.
>
> **Q:** *"Can an attacker with some knowledge about this watermarking scheme exploit the cyclic nature of the watermark? For example, can they determine the vocab set for each tag? And can they align the edits to the cycle (e.g., only edit tokens at tag 1)?.*"
>
> **A:** Thank you for raising these points. In our current implementation, sub-vocabulary partitioning is done randomly in the very beginning but fixed across generations for simplicity. While we use static partitioning for simplicity, our method is *fully compatible with dynamic partitioning variants*. For example, a keyed hash over previous tokens can be used to dynamically select patterns or partitions, similar to KGW. During the edit detection phase, we can let the detector access the key and thus infer the dynamic green/red lists for detection. We view this as a natural extension to enhance security and does not impact our edit detection. We are currently running experiments with such randomly selected sub-vocabulary sets, and will update the results to the revision once they are completed.
>
> Finally, we thank the reviewer again for the thoughtful feedback and hope our responses have clarified the key points. Please let us know if any further questions arise, and we sincerely appreciate the opportunity to clarify and strengthen our submission.

---

> > ### Author Response · Authors · 2025-11-28
> >
> > Thank you again for raising the concern regarding potential vulnerabilities arising from deterministic vocabulary partitions. As noted in our first response, we have conducted **additional experiments using randomized vocabulary partitions** at each token index. These results have now been updated in the revision.
> >
> > Specifically, we added Remark 3.1 in Section 3.1 (highlighted in blue in the revision) to clarify that the vocabulary partition can be randomized at each token index to enhance robustness, similar in spirit to the hashing-based randomization used in KGW. In addition, Table 3 in the Appendix now reports numerical results for a randomized variant of our alternating binary pattern (“AB (rand)”). This variant randomizes the green/red vocabulary split at each token index $t$ based on the previous token $s^{(t-1)}$, that is, we compute a hash of the previous token, use it to seed a random number generator, and then partition the vocabulary into green and red lists according to that seed. As summarized in the updated Appendix text (also highlighted in blue), we note that randomizing the green/red vocabulary split is by definition compatible with our watermark construction, and it improves edit-detection accuracy relative to the fixed partition in several regimes.
> >
> > Overall, these new experiments demonstrate that our method can readily adopt randomized vocabulary partitions when desired, while continuing to achieve comparable or even stronger edit-detection performance. We hope these additional results can address the reviewer’s concern, and we appreciate the suggestion that motivated this natural extension.

---

### Author Response · Authors · 2025-12-03
**Author Final Comment**

We sincerely thank all reviewers for their thoughtful and constructive feedback. We are especially grateful for the shared recognition of (i) the importance and novelty of the *post-generation edit detection* task, (ii) the practicality and extensibility of our combinatorial watermarking framework, and (iii) the effectiveness of our method in detecting and localizing edits across different LLMs.

---

### Summary of Revisions and Additions
**1. Expanded numerical results and clearer evaluation (Reviewer Jmfd, Reviewer gEsA).**
We added:
- **Table 2** reporting *exact numerical values* corresponding to Figure 5;
- **Table 3** showing detection accuracy under *smaller*  $\delta$ values, demonstrating that our method retains strong performance across a wider range of watermarking strengths.

These additions make the evaluation clearer, more precise, and easier to interpret.

**2. Justification of the trade-off between text quality and edit detectability (Reviewer Jmfd, Reviewer kxy5).**
We incorporated PPL (perplexity) results into **Table 3** and added qualitative text samples in **Tables 7–8**. We further clarified in the revision that:
- the impact on text quality reflects a *tunable trade-off* between edit detectability and output quality, and
- this impact is *modest* in practice. For example, Table 3 demonstrates that, under the simplest AB pattern, PPL remains comparable to Unigram and KGW across various  $\delta$ values.

**3. Comparison with watermarked-segment detection baselines (Reviewer Jmfd).**
We added a new Related Work paragraph and new empirical evaluation (Appendix B.2, **Table 6**) comparing with:
- *[1] Efficiently Identifying Watermarked Segments in Mixed-Source Texts*, and
- *[2] WaterSeeker: Pioneering Efficient Detection of Watermarked Segments in Large Documents.*.

In summary, these segment-detection methods are designed to identify long contiguous watermarked spans and are not directly optimized when *adapted to* detecting sparse post-generation edits. In contrast, our cyclic pattern is specifically structured to flag local edits and achieves substantially higher detection accuracy, as shown in Table 6.

**4. Performance under realistic mixed edits (Reviewer Jmfd, Reviewer D4Hi, Reviewer gEsA).**
We have two results that supplement the three canonical edit types presented in the main text:
- **Gemini-generated misinformation edits** (already included in the original submission), achieving ≥81% accuracy and often exceeding 90% (Table 5),
- **LLaMA-2-13B mixed edits** (insert/delete/replace across multiple positions), achieving consistently >95% detection accuracy (newly added **Table 4**).

These results demonstrate resilience to realistic and mixed edit scenarios.

**5. Clarification of relation to multi-bit watermarking (Reviewer gEsA).**
We added a new Related Work paragraph clarifying that our combinatorial pattern is *not* equivalent to multi-bit watermarking schemes (e.g., Yoo et al., 2024). In short, Multi-bit watermarks embed arbitrary bitstrings for attribution; our method instead establishes cyclic token-level constraints tailored for edit localization.

**6. Addressing adversarial concerns with randomized partitions (Reviewer kxy5).**
We added **Remark 3.1** and new experiments with **randomized vocabulary partitions** (“AB (rand)”), which can enhance robustness compared to using a fixed partition. Results in **Table 3** show that such randomization is compatible with our framework and can yield even stronger edit-detection accuracy.

**7. Generation-time comparison (Reviewer gEsA).**
We added **Table 1**, showing negligible overhead between unwatermarked and watermarked generation (e.g., 4.22s vs. 4.33s for 2,048 tokens).

---

### Concluding Remarks

The revised manuscript now includes expanded experiments, detailed text-quality analyses, evaluations under mixed edits, and clearer and more comprehensive comparisons with prior work. We hope these revisions satisfactorily address all concerns and enhance the clarity, rigor, and completeness of our submission.

We greatly appreciate the opportunity to refine and strengthen our work.

---

### Meta-Review · Area_Chair_LBDZ · 2026-01-05

**Summary:**

Reviewers agree that detecting and localizing post-generation edits is an important and timely problem, and the paper proposes a clear combinatorial watermarking approach with strong empirical performance on token-level edits.

However, several reviewers expressed concerns that the core watermarking mechanism is largely incremental relative to prior list-based or multi-bit watermarking schemes, and that the reported gains come with nontrivial trade-offs in text quality and global watermark detectability. The AC also concurred with the assessment that the watermarking design lacks sufficient novelty.

Given the remaining questions about robustness, generality, and practical deployment, the paper does not clearly meet the ICLR acceptance bar.

**Reviewer Concerns:**

The rebuttal addressed many concrete issues, including adding additional experimental results, text-quality (PPL) evaluation, mixed-edit experiments, and comparisons to segment-detection baselines. However, core concerns remain regarding vulnerability to knowledgeable or adaptive adversaries, limited robustness to semantic-preserving paraphrasing, and the incremental nature of the watermarking design itself. These remaining issues limit the strength and generality of the paper’s claims.

**Reviewer Scores:**

Reviewer Jmfd would likely remain a reject, as their central concerns about text quality impact and practical viability were partially alleviated. Reviewers kxy5, gEsA, and D4Hi would likely maintain borderline scores, as they acknowledged improvements but did not fully revise their reservations. Overall, the score distribution would remain mixed.

---

### Decision · Program_Chairs · 2026-01-26

Reject